# Bethe Ansatz, Quantum Circuits, and the F-basis

**Roberto Ruiz**[1a], **Alejandro Sopena**[2a], **Esperanza López**[3a], **Germán Sierra**[4a],
**Balázs Pozsgay**[5b]

[a] Instituto de Física Teórica UAM/CSIC
Universidad Autónoma de Madrid
C/ Nicolás Cabrera 13–15
Cantoblanco, 28049 Madrid, Spain

[b] MTA-ELTE "Momentum" Integrable Quantum Dynamics Research Group,
ELTE Eötvös Loránd University
Pázmány Péter sétány 1/a,
1117 Budapest, Hungary

The Bethe Ansatz is a method for constructing exact eigenstates of quantum-integrable spin chains. Recently, deterministic quantum algorithms, referred to as "algebraic Bethe circuits", have been developed to prepare Bethe states for the spin-1/2 XXZ model. These circuits represent a unitary formulation of the standard algebraic Bethe Ansatz, expressed using matrix-product states that act on both the spin chain and an auxiliary space. In this work, we systematize these previous results, and show that algebraic Bethe circuits can be derived by a change of basis in the auxiliary space. The new basis, identical to the "F-basis" known from the theory of quantum-integrable models, generates the linear superpositions of plane waves that are characteristic of the coordinate Bethe Ansatz. We explain this connection, highlighting that certain properties of the F-basis (namely, the exchange symmetry of the spins) are crucial for the construction of algebraic Bethe circuits. We demonstrate our approach by presenting new quantum circuits for the inhomogeneous spin-1/2 XXZ model.

---

[1] roberto.ruiz@ift.csic.es

[2] alexsopena15@gmail.com

[3] esperanza.lopez@csic.es

[4] german.sierra@csic.es

[5] pozsgay.balazs@ttk.elte.hu

# 1 Introduction

Quantum-integrable models are distinguished many-body systems in one dimension that possess a tower of commuting conserved charges [1]. The Bethe Ansatz is a method to solve quantum-integrable models that have particle conservation, providing explicit formulae for energies, eigenstates, scalar products, correlation functions, etc. The coordinate Bethe Ansatz (CBA) solves the spectral problem by using linear superpositions of plane waves, or "magnons", as trial functions [2, 3]. The algebraic Bethe Ansatz (ABA) systematizes this approach by the R-matrix and the monodromy matrix [4–6]. Both methods enable the construction of Bethe states, which are eigenstates of the Hamiltonian when their spectral parameters satisfy the Bethe equations.

The preparation of Bethe states in spin-1/2 chains has great potential in quantum computing. For instance, Bethe states can be used to initialize quantum algorithms of adiabatic [7] and real-time [8] evolution, as well as to benchmark quantum devices. Recent research focused

on the preparation of Bethe states of the paradigm of quantum-integrable spin-1/2 chain: the homogeneous XXZ model [1], whose Bethe states form a complete basis of the Hilbert space [13].

A first class of quantum algorithms [14–16] are based on the special simplicity of Bethe states in the homogeneous XXZ model, rather than quantum integrability itself. They apply in presence of closed [14, 16] and open boundary conditions [15, 16], and are either probabilistic [14, 15] or deterministic [16]. Algorithms must be efficient to be implementable, which, in the circuit model of quantum computing, means the number of one- and two-qubit gates must grow polynomially with the parameters of the circuit. The number of gates of probabilistic algorithms is polynomial in the number of qubits $N$ and magnons $M$ [14, 15]. However, the success probability decreases exponentially with $N$ for the ground state, and super-exponentially with $M$ for excited states if $N$ is large [17]. The number of gates of the deterministic algorithm in [16], related to quantum encoders [18], is linear in $N$, but exponential in $M$ again.

A second class of deterministic algorithms relying explicitly on quantum integrability goes by the name of "algebraic Bethe circuits (ABCs)" [19, 20]. Just as the ABA builds Bethe states as "creation" operators atop a (pseudo-)vacuum, ABCs seek to frame Bethe states as unitaries acting on a reference state. ABCs were proposed for the homogeneous XXZ model with periodic boundaries [19,20]. The starting point of [19] was the representation of Bethe states of the ABA as matrix-product states (MPSs) [21]. MPSs are the simplest tensor networks [22], which make the entanglement structure of states in one dimension apparent through a circuit-like arrangement of local tensors that act on both the Hilbert space of the spin chain and an auxiliary space. The unitaries of ABCs were extracted numerically from these tensors for the ABA in [19]. Closed formulae for the unitaries of ABCs were later obtained in [20] by a complementary approach. The key step in [20] was deriving a new representation of the linear superpositions of plane waves in the CBA as an MPS. The tensors of this MPS, unlike that of the ABA, directly lead to analytical expressions for the unitaries of ABCs. Nonetheless, the proof of the construction of [20] was not complete and partially relied on numerical checks that were performed for small number of magnons $M$. The equivalence between the realizations of ABCs in [19, 20], and thus between the formulations of the ABA and the CBA as MPSs, was also verified for small $M$ in [20]. The number of unitaries in ABCs is linear in the number of qubits $N$. However, the ABC unitaries act on up to $M + 1$ qubits, and the efficiency of their decomposition into one- and two-qubit gates on $M$ remains uncertain in general. The generalization of the ABC construction to open boundaries appears in [23].

The results of [20] raise the question of systematizing ABCs. A clear method for formulating Bethe states of the ABA as the circuits of [20] would enhance the search for quantum-integrable models in which the unitaries of ABCs admit efficient decompositions. The missing link preventing the systematization of ABCs is the connection between the MPS of the ABA and the CBA. The ABA, which can be straightforwardly identified with an MPS [21], is the standard method for computing Bethe states, while the MPS of the CBA provides closed formulae for the unitaries of ABCs. In this work, we propose that the change to the F-basis of [24] in the auxiliary space is the key to transforming the MPS of the ABA into that of the CBA, thereby allowing us to analytically rephrase Bethe states as quantum circuits. The crucial feature of the F-basis is that it is invariant with respect to exchange of qubits. In addition, we address the loopholes of [20] by

---

[1]The ground state of the anti-ferromagnetic spin chain has been approximated by a double-bracket quantum algorithm in [9]. Moreover, the homogeneous XXZ model has been considered in connection to the variational quantum eigensolver in [10, 11] and shows promise in sampling certain topological invariants [12].

presenting a rigorous method to eliminate auxiliary qubits in the final circuit. We illustrate our approach with new ABCs for the inhomogeneous XXZ model with periodic boundaries. Figure 1 contains an overview of the realizations of the Bethe Ansatz that we uncover.

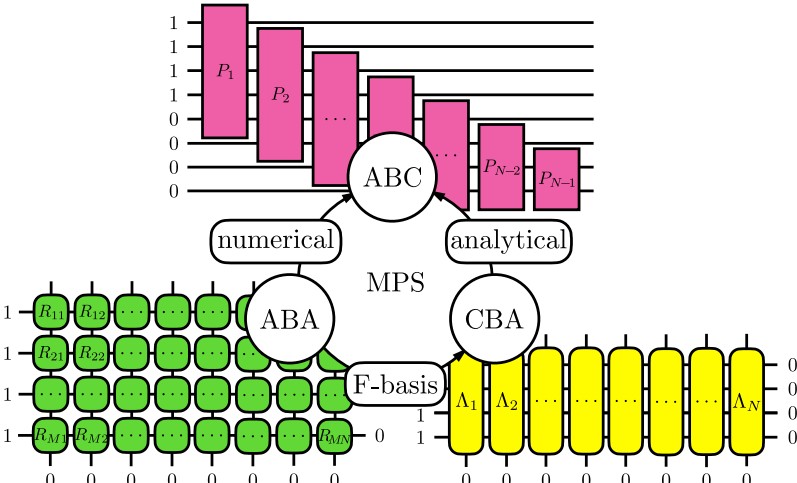

Figure 1: Diagram of realizations of the Bethe Ansatz.

The F-basis is a special basis where the operators of the ABA are local operators with a non-local dressing [24]. Originally proposed for the inhomogeneous XXZ model with periodic boundaries [24], the F-basis, which admits a diagrammatic representation [25], was soon extended to encompass higher-spin [26–28], higher-rank [29–31], totally anisotropic [32], open [33,34], and supersymmetric [35, 36] chains, among others [37, 38]. (We refer to [25, 39] for a summary of the F-basis.) The F-basis in the quantum space was instrumental in providing a first solution to the quantum inverse scattering problem for the spin operators [39], laying the groundwork for the exact computation of form factors and correlation functions. Other applications of the F-basis included the computation of domain-wall partition functions [40,41], asymmetric simple exclusion processes [42], and spin-chain propagators [43]. The importance of applying the F-basis to the MPS of the Bethe Ansatz was envisaged in [44], which reviewed the equivalence of the MPS of [45–47] with the ABA by a change of basis in the auxiliary space as presented in [48]. The same remark was made in [49], which also underscored the suitability of the F-basis for the explicit computation of Bethe wave functions.

The paper has the following structure. In Section 2, we present the inhomogeneous XXZ model with periodic boundaries, together with the monodromy matrix and the R-matrix. We introduce the F-basis in the auxiliary space and show it guarantees the invariance with respect to exchange of qubits in the auxiliary space. In Section 3, we prove the change to the F-basis in the auxiliary space relates the formulation of the ABA and the CBA as an MPS. First, we detail the connection for the homogeneous XXZ model, and elaborate on the interplay between the F-basis and Bethe states. Next, we turn to the inhomogeneous XXZ model, where we present a simple parameterization of the CBA for the inhomogeneous spin chain. In Section 3, we use the MPS of the CBA to write the unitaries of ABCs for the inhomogeneous XXZ model. We eliminate the auxiliary qubit rigorously by a suitably re-defined MPS for the smaller unitaries. In Section 4, we conclude with general remarks and prospects on future research. Appendices A–D contain demonstrations of claims in the main text and additional material.

## 2 The F-basis of the XXZ Model

In this section, we review the inhomogeneous and periodic XXZ model. In Subsection 2.1, we introduce the model, the R-matrix, and the monodromy matrix. We present the exchange algebra of the monodromy matrix, whereby the ABA follows. In subsection 2.2, we review the F-basis of [24]. We highlight that operators of the ABA in the F-basis are symmetric with respect to exchange of qubits. The property proves to be instrumental in derivation of ABCs in Section 3 into this analysis. We refer to [25, 39] for a summary of the F-basis of [24].

### 2.1 The R-matrix and Monodromy Matrix

We begin by briefly reviewing the ABA for the homogeneous XXZ model with periodic boundaries. The review serves to both provide context and facilitate the extension of the ABA to the inhomogeneous spin chain. The XXZ model is a chain of $N$ spin-1/2 sites, or, alternatively, "qubits". The Hilbert space is

$$\mathsf{H}_N = \bigotimes_{j=1}^{N} \mathsf{h}_j , \quad \mathsf{h}_j \cong \mathbb{C}^2 , \tag{2.1}$$

and goes by the name of "quantum space". We call the qubits of the quantum space "spins" to differentiate them from auxiliary qubits below. The subscript $j = 1, \ldots, N$ labels the individual Hilbert space of the spins $\mathsf{h}_j$, which is isomorphic to $\mathbb{C}^2$. The computational basis of a qubit corresponds to up and down spin-1/2 states like

$$|\uparrow\rangle := |0\rangle = \begin{bmatrix} 1 \\ 0 \end{bmatrix} , \quad |\downarrow\rangle := |1\rangle = \begin{bmatrix} 0 \\ 1 \end{bmatrix} . \tag{2.2}$$

The Hamiltonian is

$$H = \sum_{j=1}^{N} \left( X_j X_{j+1} + Y_j Y_{j+1} + \Delta Z_j Z_{j+1} \right) , \tag{2.3}$$

where $\Delta$ is the anisotropy parameter. We introduced standard Pauli matrices on the $j$-th spin subject to periodicity. To diagonalize the Hamiltonian, one considers the monodromy matrix, the $2 \times 2$-matrix whose entries are the operators of the ABA on the quantum space:

$$T(u) = \begin{bmatrix} A(u) & B(u) \\ C(u) & D(u) \end{bmatrix} \in \text{End}(\mathsf{h}_0 \otimes \mathsf{H}_N) , \quad \mathsf{h}_0 \cong \mathbb{C}^2 . \tag{2.4}$$

The variable $u$ denotes the spectral parameter. The $2 \times 2$-matrix acts on $\mathsf{h}_0$, called "auxiliary space", which corresponds to an auxiliary qubit called "ancilla". The ABA dictates the construction of eigenstates by applying one of the non-diagonal operators from the monodromy matrix to a reference state, whose spectral parameters must fulfil the Bethe equations. These Bethe states not only diagonalize the Hamiltonian, but also the transfer matrix,

$$t(u) = \text{tr}\, T(u) = A(u) + D(u) , \tag{2.5}$$

for every $u$. The regular series of the transfer matrix around every point spans a tower of commuting conserved charges diagonalized by the ABA, the standard hallmark of quantum integrability [1]. The cornerstone of the ABA is the R-matrix, which we introduce next.

The R-matrix of the XXZ model is

$$R(u) = \begin{bmatrix} 1 & 0 & 0 & 0 \\ 0 & f(u) & g(u) & 0 \\ 0 & g(u) & f(u) & 0 \\ 0 & 0 & 0 & 1 \end{bmatrix} , \quad f(u) = \frac{\sinh u}{\sinh(u + \mathrm{i}\gamma)} , \quad g(u) = \frac{\sinh(\mathrm{i}\gamma)}{\sinh(u + \mathrm{i}\gamma)} . \tag{2.6}$$

The anisotropy parameter $\Delta$ that characterizes the spin chain depends on $\gamma$ like [2]

$$\Delta = \cos \gamma . \tag{2.7}$$

The R-matrix is an operator on the Hilbert space of two qubits, and we understand $R(u)$ as a two-qubit tensor which depends on the difference of the spectral parameters of each qubit $u := u_1 - u_2$. The non-vanishing components of the R-matrix are

$$R_{00}^{00}(u) = R_{11}^{11}(u) = 1 , \quad R_{01}^{01}(u) = R_{10}^{10}(u) = f(u) , \quad R_{01}^{10}(u) = R_{10}^{01}(u) = g(u) . \tag{2.8}$$

We depict the R-matrix as a tensor in Figure 2.

$$R_{cd}^{ab}(u_1 - u_2) = c - \boxed{R} - a$$

Figure 2: The R-matrix (2.6) as a two-qubit tensor. The R-matrix acts from left to right on the first qubit, associated to $u_1$, and from bottom to top on the second, associated to $u_2$.

The R-matrix satisfies the Yang-Baxter equation (YBE) in difference form:

$$R_{12}(u_1 - u_2)R_{13}(u_1 - u_3)R_{23}(u_2 - u_3) = R_{23}(u_2 - u_3)R_{13}(u_1 - u_3)R_{12}(u_1 - u_2) . \tag{2.9}$$

Subscripts denote the Hilbert space of the pair of qubits the R-matrix acts on:

$$R_{12} = R \otimes 1_2 , \quad R_{23} = 1_2 \otimes R , \quad R_{13} = (\Pi \otimes 1) R_{23} (\Pi \otimes 1_2) , \tag{2.10}$$

where $1_2$ denotes the identity $2 \times 2$-matrix and the transposition $4 \times 4$-matrix $\Pi$ is

$$\Pi = \begin{bmatrix} 1 & 0 & 0 & 0 \\ 0 & 0 & 1 & 0 \\ 0 & 1 & 0 & 0 \\ 0 & 0 & 0 & 1 \end{bmatrix} . \tag{2.11}$$

Figure 3 depicts the YBE as an equality between tensor networks of R-matrices.

The monodromy matrix (2.4) spans an algebra with respect to the product in the auxiliary space called "exchange algebra". The algebra is associative and unital, but not commutative. The R-matrix is the intertwiner that encodes non-commutativity through the RTT-relation:

$$R_{12}(u - v)T_1(u)T_2(v) = T_2(v)T_1(u)R_{12}(u - v) , \tag{2.12}$$

[2]References [19, 20] mainly addressed the critical homogeneous XXZ model, where $-1 < \Delta \leqslant 1$ ($0 \leqslant \gamma < \pi$). Since we also consider the inhomogeneous spin chain here, whose phase diagram is not known, we allow complex $\gamma$.

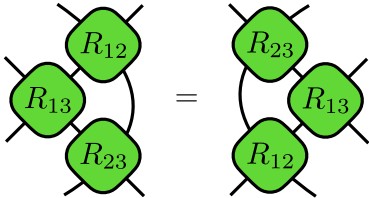

Figure 3: The YBE (2.9) as an equality of tensor networks. Each R-matrix acts on two qubits from bottom to top, and so it is read the diagram.

where

$$T_1 = T \otimes 1_2 \, , \quad T_2 = 1_2 \otimes T \, . \tag{2.13}$$

Note the R-matrix acts on the auxiliary space of two ancillae, and the monodromy matrix acts on the Hilbert space of an ancilla and on the quantum space. The RTT-relation specifies the standard commutation relations of the ABA of the XXZ model; see (1.11)–(1.24) of Chapter VII of [4]. The RTT-relation also implies that transfer matrices with different spectral parameters commute, hence they are simultaneously diagonalizable.

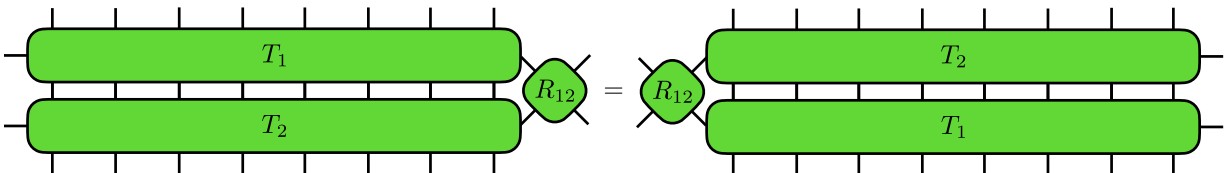

Figure 4: The RTT-relation (2.12) via tensor networks. The monodromy and R-matrices act on ancillae from left to right, The monodromy matrices act on spins from bottom to top.

The monodromy matrix of the XXZ model is a tensor network of R-matrices, where the YBE (2.9) implies the RTT-relation (2.12). Let $R_{0j} \in \mathrm{End}(\mathsf{h}_0 \otimes \mathsf{h}_j)$ be the R-matrix of the ancilla and the $j$-th spin. The monodromy matrix of the inhomogeneous XXZ model is

$$T(u) = R_{0N}(u - v_N) \ldots R_{02}(u - v_2) R_{01}(u - v_1) \, , \tag{2.14}$$

where $v_j$ is the inhomogeneity of the $j$-th spin. Figure 5 illustrates the monodromy matrix as a tensor network of R-matrices. The YBE (2.9) implies the RTT-relation (2.12). Even though (2.14) defines a one-parameter family of transfer matrices by (2.5), inhomogeneities forbid the tower of conserved charges to be local, as often required to quantum-integrable models [1]. Lacking a local Hamiltonian, we define the inhomogeneous spin chain by the transfer matrix. If $v_j = v$, the spin chain is homogeneous, and there is a tower of local conserved charges that contains the Hamiltonian (2.3). The charges are proportional to logarithmic derivates of the transfer matrix thanks to regularity $R(0) = \Pi$ [4].

We close this subsection by noting that the exchange algebra must allow for products of more than two monodromy matrices. Consider the product of three monodromy matrices. Inverting the product by alternative sequences of pair-wise swappings leads us to

$$T_1 T_2 T_3 = R_{23}^{-1} R_{13}^{-1} R_{12}^{-1} T_3 T_2 T_1 R_{12} R_{13} R_{23} = R_{12}^{-1} R_{13}^{-1} R_{23}^{-1} T_3 T_2 T_1 R_{23} R_{13} R_{12} \, , \tag{2.15}$$

where

$$R_{ab} := R_{ab}(u_a - u_b) \, , \quad T_a := T_a(u_a) \, . \tag{2.16}$$

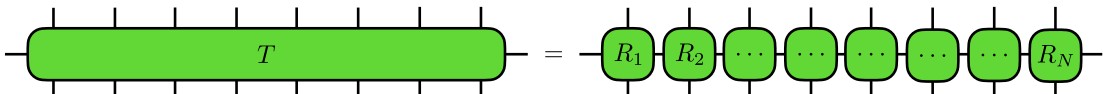

Figure 5: The monodromy matrix as a tensor network according to (2.14). The notation is $R_j := R_{0j}(u - v_j)$. The action on ancillae goes from left to right and on spins from bottom to top. The R-matrices on the diagram and the equation appear in reverse order: the leftmost $R_1$ of the tensor network acts first, the next-to-leftmost $R_2$ acts second, etc., whereas the rightmost $R_1$ in (2.14) acts first, the next-to-rightmost $R_2$ acts second, and so on.

The RTT-relation is compatible with (2.15) thanks to the YBE (2.9), in the sense that alternative sequences of transpositions giving the same product are equivalent. The YBE similarly ensures the consistency of products of a higher number of monodromy matrices. Now, the product of two monodromy matrices admits two reorderings, the trivial reordering and the transposition, each corresponding to an element of $S_2$. Conjugation by $1_2$ realizes the identity, which leaves the order unaffected. Conjugation by the R-matrix realizes the transposition, which inverts the order, in agreement with the RTT-relation (2.12),

The product of $M$ monodromy matrices admits $M!$ reorderings. Reorderings are in one-to-one correspondence with permutations $\sigma \in S_M$. Each $\sigma$ corresponds to a $2^M \times 2^M$-matrix $R^\sigma_{12\ldots M}$ on the auxiliary space of $M$ ancillae

$$\mathsf{H}_M = \bigotimes_{a=1}^{M} \mathsf{h}_a , \quad \mathsf{h}_a \cong \mathbb{C}^2 , \tag{2.17}$$

which satisfies

$$R^\sigma_{12\ldots M} T_1 T_2 \ldots T_M = T_{\sigma_1} T_{\sigma_2} \ldots T_{\sigma_M} R^\sigma_{12\ldots M} , \tag{2.18}$$

where

$$R^\sigma_{12\ldots M} := R^\sigma_{12\ldots M}(u_1, u_2, \ldots, u_M) , \tag{2.19}$$

Each $R^\sigma_{12\ldots M}$ factorizes into products of standard R-matrices. For instance, if $M = 3$, the R-matrix that performs the inversion (we use cycle notation) $\sigma = (1,3)(2)$ in (2.15) is

$$R^{(1,3)(2)}_{123} = R_{12} R_{13} R_{23} . \tag{2.20}$$

Alternative factorizations of the same R-matrix are consistent owing to the YBE (2.9). We are now in position to introduce F-matrices and the F-basis.

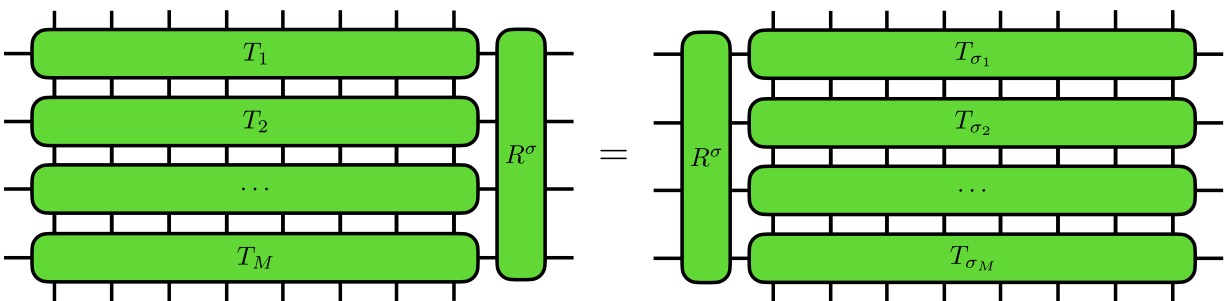

Figure 6: Permutation of the product of $M$ monodromy matrices by $R^\sigma := R^\sigma_{12\ldots M}$ in (2.18).

## 2.2 The F-basis

By definition, the F-matrix is the invertible $4 \times 4$-matrix that satisfies

$$R_{12}(u) = F_{21}^{-1}(-u) F_{12}(u) , \qquad (2.21)$$

where

$$F_{21} = \Pi \, F_{12} \, \Pi . \qquad (2.22)$$

The F-matrix of the R-matrix (2.6) is

$$F_{12}(u) = \begin{bmatrix} 1 & 0 & 0 & 0 \\ 0 & 1 & 0 & 0 \\ 0 & g(u) & f(u) & 0 \\ 0 & 0 & 0 & 1 \end{bmatrix} . \qquad (2.23)$$

The definition holds thanks to the following properties of the functions in (2.6):

$$f(u)f(-u) + g(u)g(-u) = 1 , \quad f(u)g(-u) + g(u)f(-u) = 0 . \qquad (2.24)$$

The F-matrix exists because the R-matrix is pseudo-unitary [24] [3]:

$$R_{12}(u)R_{21}(-u) = 1_4 . \qquad (2.25)$$

An F-matrix $F_{12\ldots M}$ is the $2^M \times 2^M$-matrix that encodes all the R-matrices for products of $M$ monodromy matrices. The definition of F-matrices in this case is

$$R_{12\ldots M}^{\sigma} = F_{\sigma_1 \sigma_2 \ldots \sigma_M}^{-1} F_{12\ldots M} , \qquad (2.26)$$

where

$$F_{12\ldots M} := F_{12\ldots M}(u_1, u_2, \ldots, u_M) , \quad F_{\sigma_1 \sigma_2 \ldots \sigma_M} := \Pi^{\sigma} F_{12\ldots M}(u_{\sigma_1}, u_{\sigma_2}, \ldots, u_{\sigma_M}) \Pi^{\sigma} , \qquad (2.27)$$

and $\Pi^{\sigma}$ is the permutation $2^M \times 2^M$-matrix of $\sigma$. Note that the definition of $F_{\sigma_1 \sigma_2 \ldots \sigma_M}$ involves both the permutation of ancillae by $\Pi^{\sigma}$ and the permutation of the arguments of the F-matrix. The closed formula of $F_{12\ldots M}$ is [25, 31]

$$F_{12\ldots M} = \prod_{a=1}^{M-2} \left[ |0\rangle_a \langle 0|_a + |1\rangle_a \langle 1|_a \prod_{b=a+1}^{M} R_{ab} \right] , \qquad (2.28)$$

where $|i\rangle_a \langle j|_a$ are projectors of the Hilbert space $\mathsf{h}_a$ in (2.17) and

$$R_{ab} := R_{ab}(u_a - u_b) . \qquad (2.29)$$

The F-matrices $F_{12\ldots M}$ realize the reordering of the product of monodromy by means of twists. The definition (2.26) enables us to rephrase (2.18) as

$$F_{12\ldots M} T_1 T_2 \ldots T_M F_{12\ldots M}^{-1} = F_{\sigma_1 \sigma_2 \ldots \sigma_M} T_{\sigma_1} T_{\sigma_2} \ldots T_{\sigma_M} F_{\sigma_1 \sigma_2 \ldots \sigma_M}^{-1} . \qquad (2.30)$$

---

[3]The pseudo-unitarity of the R-matrix differs from matrix unitarity in general. For instance, (2.6) is pseudo-unitary according (2.25), but unitary only if $u$ is real. We emphasize the name "unitarity" for the pseudo-unitarity of the R-matrix (2.25), which borrows from factorized-scattering theory [50], is deeply ingrained in the literature.

The consistency of the F-matrix $F_{12}$ with the algebra of the monodromy matrix implies the consistency among $F_{12...M}$ with different $M$ [24]. The F-matrices realize the change to the "F-basis" of the auxiliary space (2.17), whereby the operators of the ABA are particularly simple [24].

Before proceeding, we should make a remark. Reference [24] initially introduced F-matrices on the quantum space (2.1). The corresponding F-basis is useful for computing scalar products [24, 39] and solving the quantum inverse scattering problem for local spin operators [39]. In this work, however, we focus on F-matrices on the auxiliary space, as proposed in [43]. This approach shall prove to be well-suited for framing Bethe states as quantum circuits.

Let us consider the product $M$ monodromy matrices. Figure 7 depicts the product as a tensor network of R-matrices. Monodromy matrices admit a dual picture where spins and ancillae switch roles. (We keep the nomenclature "spins" and "ancillae" in the dual picture with the same meaning to make the context clear.) The definition of the $j$-th dual monodromy matrix over $M$ ancillae is [4]

$$\mathscr{T}_j(v_j) := \mathscr{T}_j := R_{1j}(u_1 - v_j)R_{2j}(u_2 - v_j)\dots R_{Mj}(u_M - v_j) . \tag{2.31}$$

The spectral parameter of the $j$-th matrix is $v_j$, whereas $u_a$ is the inhomogeneity of the $a$-th ancilla. The YBE (2.9) implies the RTT-relation of dual monodromy matrices is

$$\mathscr{T}_1(v_1)\mathscr{T}_2(v_2)R_{12}(v_1 - v_2) = R_{12}(v_1 - v_2)\mathscr{T}_2(v_2)\mathscr{T}_1(v_1) . \tag{2.32}$$

The operators of the ABA of dual monodromy matrices follow from the corresponding exchange algebra. Figure 8 depicts the product of dual monodromy matrices as a tensor network.

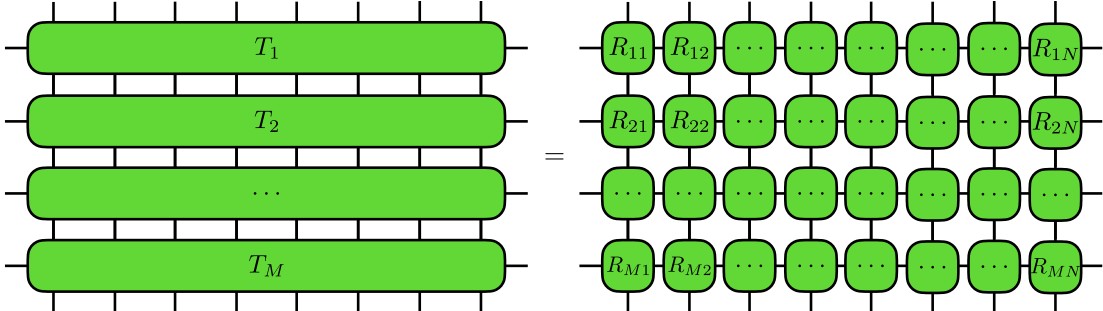

Figure 7: The product of monodromy matrices equals a tensor network of R-matrices.

To introduce the F-basis, we must perform a change of basis in the auxiliary space by the F-matrix $F_{12...M}$. According to (2.30), the product of monodromy matrices is symmetric under permutations once is twisted by the F-matrix. Here, we show that this property implies the existence of new dual monodromy matrices

$$\widetilde{\mathscr{T}_k} = F_{12...M}\mathscr{T}_k F_{12...M}^{-1} , \tag{2.33}$$

that are symmetric with respect to exchange of ancillae: [5]

$$\widetilde{\mathscr{T}_k} = \widetilde{\mathscr{T}_k}^{\sigma} , \tag{2.34}$$

---

[4] $\mathscr{T}_j$ are often called "column-to-column" monodromy matrices, as opposed to the "row-to-row" matrices $T_a$.

[5] The symmetry of monodromy matrices with respect to the exchange of spins was previously highlighted in [51]. We are grateful to A. A. Ovchinnikov for bringing this work to our attention.

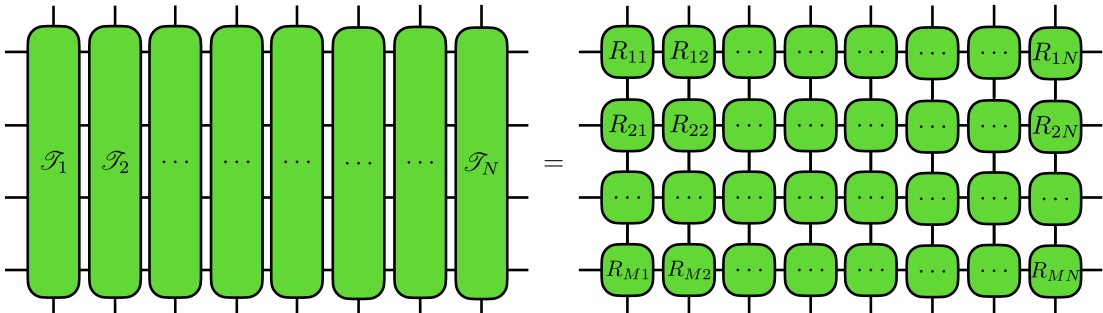

Figure 8: The product of dual monodromy matrices equals a tensor network of R-matrices. Grouping R-matrices by columns in Figure 7 leads to this picture.

where

$$\widetilde{\mathscr{T}_k} = \widetilde{\mathscr{T}_k}(v_k; u_1, \ldots, u_M) \ , \quad \widetilde{\mathscr{T}_k^\sigma} = \Pi^\sigma \widetilde{\mathscr{T}_k}(v_k; u_{\sigma_1}, \ldots, u_{\sigma_M})\Pi^\sigma \ , \tag{2.35}$$

and we wrote the dependence on $u_a$ explicitly. Figure 9 represents (2.33) and (2.34). The proof of (2.34) follows from the independence of the F-matrix on the number of spins $N$. If $N = 1$, just $\widetilde{\mathscr{T}_1}$ is defined. Equation (2.30) with $N = 1$ implies $\widetilde{\mathscr{T}_1}$ fulfills (2.34). Any other $\widetilde{\mathscr{T}_j}$ is symmetric with respect to exchange of ancillae due to the independence on $v_j$ of the proof. Moreover, we emphasize $\widetilde{\mathscr{T}_j}$ are also dual monodromy matrices: since $F_{12\ldots M}$ just acts on the auxiliary space and does not depend on $-v_j$, (2.32) holds for $\widetilde{\mathscr{T}_j}$ as well. Both $\mathscr{T}_j$ and $\widetilde{\mathscr{T}_j}$ span the same exchange algebra. The definition of $\widetilde{\mathscr{T}_j}$ in (2.33) can be understood as a change of basis of the dual monodromy matrices $\mathscr{T}_j$. The new basis is called "F-basis".



Figure 9: New dual monodromy matrices defined in (2.33). The notation is $F = F_{12\ldots M}$. These monodromy matrices are symmetric with respect to exchange of ancillae as stated in (2.34).

The operators of the ABA simplify in the F-basis. If we rearrange the tensor product on which the new dual monodromy matrices act to mimic (2.4); that is, if we write $\widetilde{\mathscr{T}_j} \in \text{End}(\mathsf{h}_j \otimes \mathsf{H}_M)$, it follows that

$$\widetilde{\mathscr{T}_j} = \begin{bmatrix} \widetilde{\mathscr{A}_j} & \widetilde{\mathscr{B}_j} \\ \widetilde{\mathscr{C}_j} & \widetilde{\mathscr{D}_j} \end{bmatrix} \ , \tag{2.36}$$

where the first three operators read [24]

$$\widetilde{\mathscr{A}_j} = \bigotimes_{a=1}^{M} \begin{bmatrix} 1 & 0 \\ 0 & f_{aj} \end{bmatrix} ,$$

$$\widetilde{\mathscr{B}_j} = \sum_{a=1}^{M} \bigotimes_{b=1}^{a-1} \begin{bmatrix} 1 & 0 \\ 0 & f_{bj}/f_{ba} \end{bmatrix} \begin{bmatrix} 0 & 0 \\ g_{aj} & 0 \end{bmatrix} \bigotimes_{c=a+1}^{M} \begin{bmatrix} 1 & 0 \\ 0 & f_{cj}/f_{ca} \end{bmatrix} , \qquad (2.37)$$

$$\widetilde{\mathscr{C}_j} = \sum_{a=1}^{M} \bigotimes_{b=1}^{a-1} \begin{bmatrix} 1/f_{ab} & 0 \\ 0 & f_{bj} \end{bmatrix} \begin{bmatrix} 0 & g_{aj} \\ 0 & 0 \end{bmatrix} \bigotimes_{c=a+1}^{M} \begin{bmatrix} 1/f_{ac} & 0 \\ 0 & f_{cj} \end{bmatrix} ,$$

with

$$f_{aj} := f(u_a - v_j) , \quad f_{ab} := f(u_a - u_b) , \quad g_{aj} := g(u_a - v_j) , \qquad (2.38)$$

while $\widetilde{\mathscr{D}_j}$ follows from the proportionality of the quantum determinant to the identity matrix [4].

## 3 The F-basis and Algebraic Bethe Circuits

In this section, we present the ABCs of [19,20] in light of the F-basis of [24]. Inspired by [20], we use the F-basis to formulate the linear superposition of plane waves in the CBA as an MPS. The tensors of this one-dimensional network inherit the symmetry with respect to exchange of qubits of the F-basis, which later plays a key role in computing analytic expressions of the unitaries of ABCs. In Subsection 3.1, we relate the MPS formulation of the CBA for the homogeneous spin chain in [20] with the F-basis. In Subsection 3.2, we use the connection to derive the CBA of the inhomogeneous XXZ model. In Subsection 3.3, we construct the ABCs for the inhomogeneous spin chain along the lines of [20].

### 3.1 The F-basis and Coordinate Bethe Ansatz: Homogeneous Spin Chain

The ABCs consist of unitaries over various qubits that act on a reference state to produce normalized Bethe states. References [19,20] obtained ABCs for the homogeneous XXZ model, whose Bethe Ansatz is well-known. The starting point of [19] was the formulation of the ABA as an MPS [21,48], whereas [20] began with a new MPS representation of the CBA. Both classes of MPSs are connected by a change of basis in the auxiliary space, which we show to correspond, essentially, to the change to the F-basis.

We begin with the unnormalized Bethe state of $M$ magnons over $N$ spins as per the ABA [4]:

$$B(u_1) \ldots B(u_M) |0\rangle^{\otimes N} = \langle 0|_1 T_1 |1\rangle_1 \ldots \langle 0|_M T_M |1\rangle_M |0\rangle^{\otimes N} , \qquad (3.1)$$

where $B(u_a)$ are the operators in (2.4), which commute among themselves, $|i\rangle_a$ belongs to $\mathsf{h}_a$ in (2.17), and $|0\rangle^{\otimes N}$ is the reference state in the quantum space. The so-called "magnons" are to be identified with plane waves, as the CBA makes clear. (See (3.10) below.) According to (2.33), we can write

$$B(u_1) \ldots B(u_M) |0\rangle^{\otimes N} = \sum_{i_j=0,1} \langle 0|^{\otimes M} \widetilde{\mathscr{T}}_N^{i_N} \ldots \widetilde{\mathscr{T}}_2^{i_2} \widetilde{\mathscr{T}}_1^{i_1} |1\rangle^{\otimes M} |i_1 \ldots i_N\rangle , \qquad (3.2)$$

where $|i\rangle^{\otimes M}$ belongs to the auxiliary space, and we introduced the following notation for the first column of (2.36):

$$\widetilde{\mathcal{T}}_j^0 := \widetilde{\mathcal{A}_j} , \quad \widetilde{\mathcal{T}}_j^1 = \widetilde{\mathcal{C}_j} . \tag{3.3}$$

To write (3.2), we also used

$$F_{12...M} |i\rangle^{\otimes M} = |i\rangle^{\otimes M} . \tag{3.4}$$

Figure 10 depicts the Bethe state according to the ABA. The Bethe state (3.2) is an MPS, whose tensors are $\widetilde{\mathcal{T}}_j^i$ and bond dimension $2^M$. MPSs represent a highly structured form to describe many-body states in one-dimension [22].

MPSs reframe wave functions through a sequence of matrices defined over an auxiliary space, which define local tensors, each associated with a spin, such that the global wave function emerges as a product of these spin-specific matrices. The dimension of the matrices connecting adjacent spins, known as the "bond dimension", quantifies the entanglement between bipartitions of the state across the interstice between spins. Nevertheless, the MPS representation of a given state is inherently non-unique. Matrices at individual spins can be modified by introducing invertible matrices in the auxiliary space between them, a transformation that leaves the overall wave function unchanged. This ambiguity is due to the fact that these insertions cancel out in the complete tensor contraction. For instance, let $V_j$ be the invertible $2^M \times 2^M$-matrices of a gauge transformation acting on the auxiliary space of (3.2). The mapping

$$\widetilde{\mathcal{T}}_j^i \mapsto V_j^{-1} \widetilde{\mathcal{T}}_j^i V_{j-1} , \quad \langle 0|^{\otimes M} \mapsto \langle 0|^{\otimes M} V_N , \quad |1\rangle^{\otimes M} \mapsto V_0^{-1} |1\rangle^{\otimes M} , \tag{3.5}$$

yields another admissible MPS representation of the Bethe state (3.2). If all the matrices of the gauge transformation are equal, the transformation is called "global", otherwise it is called "local".

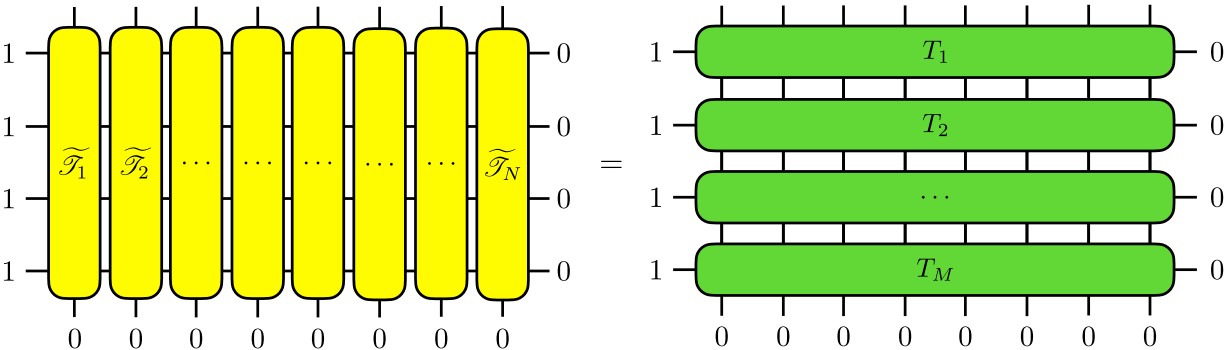

Figure 10: Bethe state according to the ABA. The standard formulation of the ABA (3.1) based on $T_a$ equals the dual formulation of the ABA in the F-basis (3.2) based on $\widetilde{\mathcal{T}}_j$.

The discussion up to this point applies to both the homogeneous and inhomogeneous XXZ models. To align with [20], whose spin chain is homogeneous, we set $v_j = 0$, hence $\widetilde{\mathcal{T}}_j^i = \widetilde{\mathcal{T}}^i$ until the end of the subsection. The MPS of the Bethe state thus becomes uniform, which means that all the tensors of (3.2) are equal.

Let us derive the representation of the CBA as an MPS from (3.2). We perform a global gauge transformation by the $2^M \times 2^M$-matrix

$$V = \bigotimes_{a=1}^M \begin{bmatrix} g_a & 0 \\ 0 & \prod_{b=1,\ b\neq a}^M f_{ab} \end{bmatrix} , \tag{3.6}$$

where

$$f_a := f(u_a) , \quad g_a := g(u_a) . \tag{3.7}$$

(We introduced $f_a$ already for convenience.) The global gauge transformation just rescales $\widetilde{\mathscr{T}}^1$ because $\widetilde{\mathscr{T}}^0$ is diagonal. The tensors of the MPS representation of (3.2) become

$$\Lambda^0 = V^{-1}\widetilde{\mathscr{T}}^0 V = \bigotimes_{a=1}^{M} \begin{bmatrix} 1 & 0 \\ 0 & f_a \end{bmatrix} ,$$

$$\Lambda^1 = V^{-1}\widetilde{\mathscr{T}}^1 V = \sum_{a=1}^{M} \bigotimes_{b=1}^{a-1} \begin{bmatrix} 1 & 0 \\ 0 & f_{ab}f_b \end{bmatrix} \begin{bmatrix} 0 & 1 \\ 0 & 0 \end{bmatrix} \bigotimes_{c=a+1}^{M} \begin{bmatrix} 1 & 0 \\ 0 & f_{ac}f_c \end{bmatrix} . \tag{3.8}$$

The transformation leaves a multiplicative factor due to the action of $V^{-1}$ and $V$ on $|0\rangle^{\otimes M}$ and $|1\rangle^{\otimes M}$, respectively. If we re-define

$$|\Psi_N^{[M]}\rangle = \left[ \prod_{a=1}^{M} \frac{1}{g_a} \prod_{\substack{b=1 \\ b\neq a}}^{M} f_{ab} \right] B(u_1)\dots B(u_M) |0\rangle^{\otimes N} , \tag{3.9}$$

we obtain

$$|\Psi_N^{[M]}\rangle = \sum_{i_j=0,1} \langle 0|^{\otimes M} \Lambda^{i_N}\dots\Lambda^{i_2}\Lambda^{i_1} |1\rangle^{\otimes M} |i_1 i_2 \dots i_N\rangle \tag{3.10}$$

$$= \sum_{1\leqslant n_1<\dots<n_M\leqslant N} \sum_{\substack{a_1,\dots,a_M=1 \\ a_p\neq a_q}}^{M} \left[ \prod_{1\leqslant q<p\leqslant M} s_{a_q a_p} \right] \left[ \prod_{p=1}^{M} x_{a_p}^{n_p-1} \right] |n_1\dots n_M\rangle_N ,$$

where

$$|n_1\dots n_M\rangle_N = \sigma_{n_1}^-\dots\sigma_{n_M}^- |0\rangle^{\otimes N} . \tag{3.11}$$

The diagonal elements of the R-matrix (2.6) define the quasi-momenta $p_1,\dots,p_M$ of the magnons [19, 20]:

$$x_a = \exp(\mathrm{i}p_a) := f_a = \frac{\sinh u_a}{\sinh(u_a + \mathrm{i}\gamma)} . \tag{3.12}$$

The scattering amplitudes are

$$s_{ab} := f_{ab} = \frac{\sinh(u_a - u_b)}{\sinh(u_a - u_b + \mathrm{i}\gamma)} , \tag{3.13}$$

in terms of which the two-body S-matrix reads

$$S_{ab} = \frac{s_{ba}}{s_{ab}} = -\frac{\sinh(u_a - u_b + \mathrm{i}\gamma)}{\sinh(u_b - u_a + \mathrm{i}\gamma)} . \tag{3.14}$$

Since the Bethe state (3.10) realizes the unnormalized linear superposition of $M$ magnons with quasi-momenta $p_1, p_2 \dots, p_M$ over $N$ spins, we call it the "MPS of the CBA".

Reference [20] used an equivalent MPS of the CBA. We prove the equivalence between both representations in Appendix A. By changing its initialization in the auxiliary space, the MPS of [20] produced Bethe states with $0 \leqslant r \leqslant M$ magnons that can have support on $1 \leqslant k \leqslant N$ qubits. This fact underpinned the construction of the unitaries of ABCs in [20]. The MPS (3.10)

also permits to define these Bethe states in the same fashion. A Bethe state of $r$ magnons over $k$ spins with quasi-momenta $p_{m_a}$ chosen among $p_a$ is

$$|\Psi_k^{[r]}\rangle = \sum_{i_j=0,1} \langle 0|^{\otimes M} \Lambda^{i_k} \ldots \Lambda^{i_2} \Lambda^{i_1} |m_1 \ldots m_r\rangle_M |i_1 \ldots i_k\rangle \qquad (3.15)$$

$$= \sum_{1 \leqslant n_1 < \ldots < n_r \leqslant k} \sum_{\substack{a_1,\ldots,a_r=1 \\ a_p \neq a_q}}^{r} \left[ \prod_{1 \leqslant q < p \leqslant M} s_{m_{a_q} m_{a_p}} \right] \left[ \prod_{p=1}^{M} x_{m_{a_p}}^{n_p-1} \right] |n_1 \ldots n_r\rangle_k \ .$$

Equation (3.15) states that the initialization of the MPS of the CBA $|m_1 m_2 \ldots m_r\rangle_M$ inside the auxiliary space produces a Bethe state whose quasi-momenta are determined by the initial state for every number of tensors $k$ of the MPS. The correspondence between Bethe states and elements of the computational basis of the auxiliary space is one-to-one for fixed $k$. We also note that (3.15) does not hold for the MPS representation of the ABA written in terms of $\mathscr{T}_j$. The demonstration of (3.15) follows from a direct computation, but there is an alternative derivation of this property. We end this subsection by proving that (3.15) is a consequence of the symmetry under exchange of ancillae of dual monodromy matrices in the F-basis (2.34). The proof also highlights the convenience of the change of normalization (3.8), as different Bethe states would carry different normalizations if defined by the MPS of $\widetilde{\mathscr{T}}^i$.

Consider (2.34). The gauge-transformation matrix (3.6) is diagonal. The effect of the gauge transformation on the matrix elements of $\widetilde{\mathscr{T}}^i$ is just a rescaling, as we already mentioned. The rescaling does not spoil the symmetry under exchange of ancillae, which still holds for (3.8):

$$\langle i_1 \ldots i_M| \Lambda^i(u_1,\ldots,u_M) |j_1 \ldots j_M\rangle = \langle i_{\sigma_1} \ldots i_{\sigma_M}| \Lambda^i(u_{\sigma_1},\ldots,u_{\sigma_M}) |j_{\sigma_1} \ldots j_{\sigma_M}\rangle \ , \qquad (3.16)$$

for every permutation $\sigma \in S_M$. It is clear from (3.8) that $\Lambda^i$ only have non-trivial matrix elements between states of the form $|m_1 m_2 \ldots m_r\rangle_M$ and $|n_1 n_2 \ldots n_{r-i}\rangle_M$. Furthermore, the contribution to the matrix elements between ancilla at $|0\rangle$ is a multiplicative factor of one. This feature, which differentiates $\Lambda^i$ from $\widetilde{\mathscr{T}}^i$, enables the reduction of the number of ancilla in the auxiliary space. We can write the action of the first $\Lambda^i$ on the initial state of (3.15) like

$$\Lambda^0(u_1,\ldots,u_M)|m_1 \ldots m_r\rangle_M = \langle 1|^{\otimes r} \Lambda^0(u_{m_1},\ldots,u_{m_r})|1\rangle^{\otimes r}|m_1 \ldots m_r\rangle_M \ , \qquad (3.17)$$

and

$$\Lambda^1(u_1,\ldots,u_M)|m_1 \ldots m_r\rangle_M = \sum_{a=1}^{r} \langle 1|^{\otimes r} \sigma_a^- \Lambda^0(u_{m_1},\ldots,u_{m_r})|1\rangle^{\otimes r} \sigma_{m_a}^+|m_1 \ldots m_r\rangle_M \ , \qquad (3.18)$$

where we used (3.16) with a permutation $\sigma$ that verifies

$$\sigma_a = m_a \ , \quad a = 1,2,\ldots,r \ , \qquad (3.19)$$

but is otherwise arbitrary. We depict (3.17) and (3.18) in Figure 11. The concatenation of (3.17) and (3.18) in the Bethe wave function of (3.15) leads us to

$$\langle 0|^{\otimes M} \Lambda^{i_k}(u_1,\ldots,u_M) \ldots \Lambda^{i_1}(u_1,\ldots,u_M)|m_1 \ldots m_r\rangle_M$$
$$= \langle 0|^{\otimes r} \Lambda^{i_k}(u_{m_1},\ldots,u_{m_r}) \ldots \Lambda^{i_1}(u_{m_1},\ldots,u_{m_r}) |1\rangle^{\otimes r} \ . \qquad (3.20)$$

Figure 11: Application of the symmetry with respect to exchange of ancillae to $\Lambda^i$. Black and red lines represent qubits at $|0\rangle$ and $|1\rangle$, respectively. Horizontal and vertical lines correspond to spins and ancillae, respectively. Numbers besides horizontal lines denote the $u_a$ identifying the ancillae.

Therefore, if we reorder the quasi-momenta in $\Lambda^i$ and trace ancillae that remain at $|0\rangle$ out, we can prove that Bethe states with a few magnons have the form (3.10). This fact is a consequence of the definitorial property of the F-basis (2.34).

Given an MPS with the property (3.15), one can construct the unitaries of ABCs in the scheme of [20]. Motivated by the connection between the CBA and the F-basis we uncovered, we now turn to the inhomogeneous spin chain to construct their ABCs.

## 3.2 The F-basis and Coordinate Bethe Ansatz: Inhomogeneous Spin Chain

In this subsection, we compute the MPS formulation of the CBA for the inhomogeneous XXZ model. This representation leads us to the exact unitaries producing normalized Bethe states in Subsection 3.3.

Let us turn back to (2.36). The formula says Bethe states of the ABA are MPSs whose tensors are $\widetilde{\mathscr{T}}^i$. In Subsection 3.1, we performed a global gauge transformation to obtain an alternative tensor (3.8) for $v_j = 0$, when the MPS is uniform and the spin chain homogeneous. To obtain analogous tensors for general $v_j$, we promote (3.6) to a local gauge transformation by means of the $2^M \times 2^M$-matrices

$$V_j = \bigotimes_{a=1}^{M} \begin{bmatrix} g_{aj} & 0 \\ 0 & \prod_{b=1, \ b\neq a}^{M} f_{ab} \end{bmatrix} . \tag{3.21}$$

The transformation acts on the tensors like

$$\begin{aligned}
\Lambda_j^0 &= V_j^{-1} \widetilde{\mathscr{T}}_j^0 V_{j-1} = \bigotimes_{a=1}^{M} \begin{bmatrix} 1 & 0 \\ 0 & f_{aj} \end{bmatrix} , \\
\Lambda_j^1 &= V_j^{-1} \widetilde{\mathscr{T}}_j^1 V_{j-1} = \sum_{a=1}^{M} \bigotimes_{b=1}^{a-1} \begin{bmatrix} 1 & 0 \\ 0 & f_{ab}f_{bj} \end{bmatrix} \begin{bmatrix} 0 & 1 \\ 0 & 0 \end{bmatrix} \bigotimes_{c=a+1}^{M} \begin{bmatrix} 1 & 0 \\ 0 & f_{ac}f_{cj} \end{bmatrix} .
\end{aligned} \tag{3.22}$$

It is convenient to assemble both operators into the non-unitary tensor

$$\Lambda_j = \Lambda_j(v_j; u_1, \ldots, u_M) : \mathsf{H}_M \otimes \mathsf{h}_j \cong \mathsf{H}_{M+1} \to \mathsf{H}_{M+1} \cong \mathsf{h}_j \otimes \mathsf{H}_M , \quad \Lambda_j^i := \langle i|_j \Lambda_j |0\rangle_j , \quad (3.23)$$

where $|i\rangle_j$ belongs to the Hilbert space of the $j$-th spin. We depict (3.22) in Figure 12. We have defined $\Lambda_j$ so that it moves the position of $\mathsf{h}_j$ in the tensor product from the last to the first place. The swapping is convenient for the construction of ABCs. We have not specified $\Lambda_j|1\rangle_j$ as it plays no role in the MPS.



Figure 12: Tensor of the non-uniform MPS of the inhomogeneous CBA from the dual monodromy matrices in the F-basis according to (3.22). The notation is $V_j = \bigotimes_{a=1}^{M} V_{j,a}$. Being factorized, the gauge transformation is a rescaling, but local rather that global due to the inhomogeneities.

The action of $V_0$ on $|1\rangle^{\otimes M}$ and $V_{N-1}^{-1}$ on $|0\rangle^{\otimes M}$ produces an overall normalization that we cancel by redefining the Bethe state like

$$|\Psi_N^{[M]}\rangle = \left[\prod_{a=1}^{M} \frac{1}{g_{aN-1}} \prod_{\substack{b=1 \\ b\neq a}}^{M} f_{ab}\right] B(u_1)\dots B(u_M) |0\rangle^{\otimes N} \ . \tag{3.24}$$

Explicitly,

$$|\Psi_N^{[M]}\rangle = \sum_{i_j=0,1} \langle 0|^{\otimes M} \Lambda_N^{i_N} \dots \Lambda_1^{i_1} |1\rangle^{\otimes M} |i_1 \dots i_N\rangle \tag{3.25}$$

$$= \sum_{1\leqslant n_1<\dots<n_M\leqslant N} \sum_{\substack{a_1,\dots,a_M=1 \\ a_p\neq a_q}}^{M} \left[\prod_{1\leqslant q<p\leqslant M} s_{a_q a_p}\right] \left[\prod_{p=1}^{M}\prod_{j=1}^{n_p-1} x_{a_p,j}\right] |n_1 \dots n_M\rangle_N \ .$$

The lack of uniformity of the MPS is reflected in the appearance of position-dependent quasi-momenta:

$$x_{a,j} = \exp(\mathrm{i}p_{a,j}) := f_{aj} = \frac{\sinh(u_a - v_j)}{\sinh(u_a - v_j + \mathrm{i}\gamma)} \ . \tag{3.26}$$

We assume $u_a$, hence $p_{a,j}$ do not satisfy the Bethe equations in principle. We assume on top that $u_a \neq u_b$: quasi-momenta are different position-wise, and the Bethe wave function does to vanish identically. The scattering amplitudes are unchanged with respect to the homogeneous spin chain and equal (3.13). We depict (3.25) in Figure 13. Since (3.25) realizes a Bethe wave functions with $M$ magnons propagating over $N$ spins the inhomogeneous spin chain, we identify it the "MPS of the CBA". Magnons are still plane waves whose scattering is governed by the two-body S-matrix (3.14), but their quasi-momenta vary as the plane wave propagates through spin chain. This parameterization straightforwardly generalizes the CBA for the homogeneous spin chain. For instance, if $M = 2$:

$$|\Psi_N^{[2]}\rangle = \sum_{1\leqslant n_1<n_2\leqslant N} \left(s_{12}\left[\prod_{j=1}^{n_1-1} x_{1,j}\right]\left[\prod_{k=1}^{n_2-1} x_{2,k}\right] + s_{21}\left[\prod_{j=1}^{n_1-1} x_{2,j}\right]\left[\prod_{k=1}^{n_2-1} x_{1,k}\right]\right) |n_1 n_2\rangle \ . \tag{3.27}$$

The proof of the equivalence between both representations appears in Appendix A.

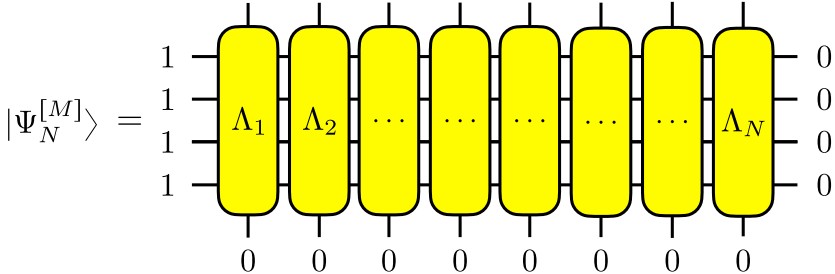

Figure 13: Bethe state as the non-uniform MPS of the CBA in the inhomogeneous spin chain. The Bethe state is a linear superposition of plane waves with spin-dependent quasi-momenta.

Like (3.10), the MPS of the CBA (3.25) enables the systematic construction of Bethe with less than $M$ magnons [6]. This fact is the upshot of (2.34), the symmetry with respect to the exchange of ancillae of $\widetilde{\mathscr{T}}_j$. Following the steps of Subsection 3.1, we indeed deduce

$$
\begin{aligned}
\langle 0|^{\otimes M} \Lambda_k^{i_k}(v_k; u_1, \ldots, u_M) \ldots \Lambda_1^{i_1}(v_1; u_1, \ldots, u_M) | m_1 \ldots m_r \rangle_M & \\
= \langle 0|^{\otimes r} \Lambda_k^{i_k}(v_k; u_{m_1}, \ldots, u_{m_r}) \ldots \Lambda_1^{i_1}(v_1; u_{m_1}, \ldots, u_{m_r}) |1\rangle^{\otimes r} & .
\end{aligned}
\tag{3.28}
$$

We should mention [49] already stressed the suitability of the F-basis to compute Bethe wave functions. We clarify the relation between (3.25) and the parameterizations of the Bethe wave functions for the inhomogeneous spin chain of [49] in Appendix A.

## 3.3 Inhomogeneous Algebraic Bethe Circuits

Having obtained the MPS of the CBA (3.25), we are in position to construct ABCs for the inhomogeneous spin chain. The first step is the decomposition the Hilbert space of $k$ qubits into eigenspaces of definite total spin along the $z$-axis:

$$
\mathsf{H}_k = \bigoplus_{r=0}^{k} \mathsf{H}_k^{[r]} ,
\tag{3.29}
$$

with

$$
\mathsf{H}_k^{[r]} = \mathrm{span} \left\{ |i_1 \ldots i_k\rangle \mid \sum_{j=1}^{k} i_j = r \right\} = \mathrm{span} \left\{ |n_1 \ldots n_r\rangle_k \mid 1 \leqslant n_1 < \ldots < n_r \leqslant k \right\} .
\tag{3.30}
$$

The dimension of the eigenspace is

$$
\dim \mathsf{H}_k^{[r]} = \binom{k}{r} ,
\tag{3.31}
$$

where we define the binomial coefficient to vanish if $r < 0$ or $r > k$. We stress (3.30) encompasses the quantum (2.1) and auxiliary (2.17) Hilbert spaces for $k = N$ and $k = M$, respectively. We introduced the decomposition because Bethe states arrange into the eigenspaces, thus the unitaries of ABCs based on them break into blocks of definite total spin along the $z$-axis. Bethe states with $r$ magnons over $k$ spins belong to $\mathsf{H}_k^{[r]}$ in particular. If $k$ and $r$ with $r \leqslant k$ are fixed, Bethe states are in one-to-one correspondence with the computational basis of $\mathsf{H}_k^{[r]}$. In

---

[6]The MPS in terms of $\widetilde{\mathscr{T}}_j^i$ (3.2) also gives rise to Bethe states with a few magnons thanks to the symmetry (2.34). However, normalizations are more intricate, which complicates the later derivation of unitaries.

Appendix B, we define the index $\alpha$ to label the string $1 \leqslant n_1 < \ldots < n_r \leqslant k$ inside the eigenspace $\mathsf{H}_k^{[r]}$. We shall use the index to label to the objects of ABCs. Later we do not allow $r$ to be greater than $M$, although $M$ itself could belong to $0 \leqslant M \leqslant N$.

The arrangement of Bethe states into eigenspaces is not accidental. In the MPS formulation of the CBA (3.25), it follows from the commutativity between $\Lambda_j$ and the total spin along $z$-axis of $M + 1$ qubits ($M$ ancillae and the $j$-th spin of the quantum space). Therefore, we can write

$$\Lambda_j^i = \bigoplus_{r=0}^{M} \Lambda_j^{[i,r]} \ , \quad \Lambda^{[i,r]} : \mathsf{H}_M^{[r]} \to \mathsf{H}_M^{[r-i]} \ . \tag{3.32}$$

The number of rows and columns of the non-unitary matrices is

$$\# \text{ rows} \times \# \text{ columns of } \Lambda_j^{[i,r]} = \binom{M}{r-i} \times \binom{M}{r} \ . \tag{3.33}$$

The square block $\Lambda^{[0,r]}$ is diagonal with entries

$$\Lambda_{j,\alpha\beta}^{[0,r]} = \delta_{\alpha\beta} \prod_{p=1}^{r} x_{n_p,j} \ , \tag{3.34}$$

where we resorted to the identification

$$|\alpha\rangle = |m_1 \ldots m_r\rangle_k \ , \quad |\beta\rangle = |n_1 \ldots n_r\rangle_k \ , \tag{3.35}$$

with $k = M$, according to Appendix B. We shall use the identification (3.35) consistently in the following. The rectangular block $\Lambda^{[1,r]}$ has the entries

$$\Lambda_{j,\alpha_r\beta}^{[1,r]} = \sum_{p=1}^{r} \delta_{\alpha_r\beta_p} \prod_{\substack{q=1 \\ q \neq p}}^{r} s_{n_p n_q} x_{n_q,j} \ , \tag{3.36}$$

where we identified

$$|\alpha_r\rangle = |m_1 \ldots \ldots m_{r-1}\rangle_{k-1} \ , \quad |\beta_p\rangle = |n_1 \ldots n_{p-1} n_{p+1} \ldots n_r\rangle_k \ , \tag{3.37}$$

with $k = M$. Given the block decomposition of $\Lambda_j$, we seek to write the MPS of the CBA (3.25) as a quantum circuit. In other words, we want to compute unitaries out of the non-unitary tensors (3.23), which themselves decompose into unitary blocks of definite total spin along the $z$-axis. We achieve this goal by a local gauge transformation that puts (3.25) into the canonical form. The canonical form is a standard representation of an MPS where the tensor is subject to orthonormalization constraints [22]. The tensor thus obtained is unique up to unitary rotations in the auxiliary space. Here we use specifically the left canonical form of the MPS (3.25), defined as that whose tensors $\Lambda_j^i$ are isometries with more rows than columns.

The set of matrices $X_j$ realize the local gauge transformation on the auxiliary space we want. The $M + 1$-qubit unitary is built as

$$P_j = X_{j+1}^{-1} \Lambda_j X_j \ . \tag{3.38}$$

We depict (3.38) in Figure 14. Unitarity holds if

$$P_j^\dagger P_j = 1_{2^{M+1}} \ . \tag{3.39}$$

We assume $X_j$ preserve the total spin along the $z$-axis,

$$X_j = \bigoplus_{r=0}^{M+1} X_j^{[r]} \,, \tag{3.40}$$

hence $P_j$ decomposes into unitary blocks according to $r$.

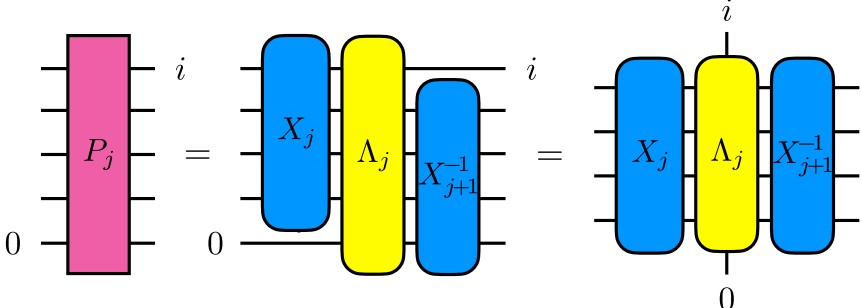

Figure 14: $M + 1$-qubit unitaries (sharp-cornered) from the non-unitary tensor (rounded-cornered) by a gauge transformation as per (3.38). Locating ancillae and spins in a single array is necessary to obtain ABCs, whose unitaries follow from the elimination of post-selected qubits.

The local gauge transformation (3.38) leads us to a quantum circuit. However, the quantum algorithm thus obtained is probabilistic. The last $M$ out of the $N + M$ qubits must be post-selected at $|0\rangle^{\otimes M}$. To avoid the computational cost of post-selection, we trace these qubits out. The result is the circuit of ABCs, a class of deterministic quantum algorithms. We still denote the unitaries by $P_j$, despite that the elimination of post-selected qubits reduces the size of the last unitaries in the circuit. ABCs produce normalized Bethe states of $M$ magnons over $N$ spins:

$$|\Phi_N^{[M]}\rangle = \frac{1}{\sqrt{\langle \Psi_N^{[M]} | \Psi_N^{[M]} \rangle}} |\Psi_N^{[M]}\rangle = P_{N-1} \dots P_2 P_1 |1\rangle^{\otimes M} |0\rangle^{\otimes N-M} \,. \tag{3.41}$$

We depict the circuit in Figure 15.

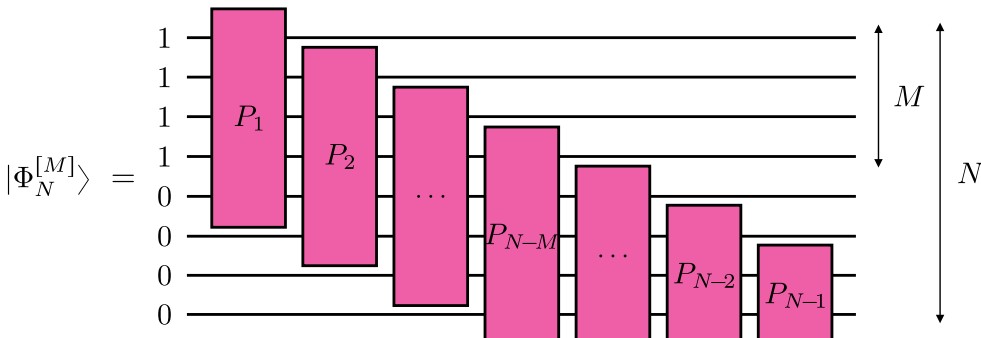

Figure 15: ABCs are deterministic algorithms preparing normalized Bethe states of the inhomogeneous XXZ model. The unitaries are either long or short.

The number of qubits on which the unitaries act defines two classes: long ($1 \leqslant j \leqslant N - M$) and short ($N - M < j \leqslant N$) unitaries. The unitaries of both classes are the orthogonal sum of unitary blocks, which in turn split into non-unitary building blocks. Long unitaries are

$$P_j = \bigoplus_{r=0}^{M+1} P_j^{[r]} \,, \quad P_j^{[i,r]} = \langle i|_j \, P_j^{[r]} \, |0\rangle_{j+M} \,, \quad 1 \leqslant j \leqslant N - M \,; \tag{3.42}$$

whereas short unitaries are

$$P_j = \bigoplus_{r=0}^{N-j+1} P_j^{[r]} \ , \quad P_j^{[i,r]} = \langle i|_j \, P_j^{[r]} \ , \quad N - M < j \leqslant N - 1 \ . \tag{3.43}$$

We summarize the properties of long and short unitaries in Table 1.

| $P_j^{[i,r]}$ | position | domain | image | # rows × # columns | formulae |
|---|---|---|---|---|---|
| long | $1 \leqslant j \leqslant N - M$ | $\mathsf{H}_M^{[r]}$ | $\mathsf{H}_M^{[r-i]}$ | $\binom{M}{r-i} \times \binom{M}{r}$ | (3.44) |
| short | $N - M < j \leqslant N - 1$ | $\mathsf{H}_{N-j+1}^{[r]}$ | $\mathsf{H}_{N-j}^{[r-i]}$ | $\binom{N-j}{r-i} \times \binom{N-j+1}{r}$ | (3.71) |

Table 1: Features of the non-unitary building blocks $P_j^{[i,r]}$ of the unitaries of ABCs.

Let us now write the exact form of the non-unitary building blocks of ABCs. The realization of Bethe states as the MPS of the CBA (3.25), which provides the means to construct Bethe states with $0 \leqslant r \leqslant M$ magnons, enables us to compute explicit expressions. We begin with long unitaries in Sub-subsection 3.3.1, which are simpler. In Sub-subsection 3.3.2, we focus on short unitaries, where we show how to eliminate the qubits post-selected at $|0\rangle^{\otimes M}$ by defining a new short tensor in the MPS. Our approach to short unitaries detaches from that of [20]. We prove the equivalence between both points of view in Appendix C.

### 3.3.1 Long Unitaries

Formula (3.38) implies the non-unitary building blocks of long unitaries are

$$P_j^{[i,r]} = X_{j+1}^{-1[r-i]} \, \Lambda_j^{[i,r]} \, X_j^{[r]} \ , \tag{3.44}$$

because they are unaffected by the elimination of ancillae. The matrix elements of $\Lambda_j$ are (3.34) and (3.36) we already introduced. To write the formulae of $X_j$ and $X_j^{-1}$, we need Bethe states with $r$ magnons over $k$ spins. We choose that the states have support in the last spins of the spin chain in view of the architecture of ABCs. The choice means Bethe states belong to

$$\mathsf{H}_k = \bigoplus_{\ell=j_k}^{N} \mathsf{h}_\ell \ , \tag{3.45}$$

where

$$j_k := N - k + 1 \ . \tag{3.46}$$

According to (3.25), the Bethe state with quasi-momenta $p_{m_a,\ell}$ out of $p_{1,\ell}, \dots, p_{M,\ell}$ is

$$|\Psi_{k,\alpha}^{[r]}\rangle = \sum_{i_\ell=0,1} \langle 0|^{\otimes M} \Lambda_N^{i_N} \dots \Lambda_{j_k}^{i_{j_k}} |m_1 \dots m_r\rangle_M |i_{j_k} \dots i_N\rangle_k \tag{3.47}$$

$$= \sum_{\substack{j_k \leqslant n_1 < \dots < n_r \leqslant N \\ a_p \neq a_q}} \sum_{\substack{a_1,\dots,a_r=1}}^{r} \left[ \prod_{1 \leqslant q < p \leqslant r} s_{m_{a_q} m_{a_p}} \right] \left[ \prod_{p=1}^{r} \prod_{\ell=j_k}^{n_p-1} x_{m_{a_p},\ell} \right] |n_1 \dots n_r\rangle_k \ .$$

Since we focus on long unitaries, $1 \leqslant j_k \leqslant N - M$, hence $M + 1 \leqslant k \leqslant N$. The number of magnons $r$ is always smaller than the number of spins $k$ over which they propagate.

Bethe states are in one-to-one correspondence with the computational basis of $\mathsf{H}_k^{[r]}$. This fact permits us to understand the MPS of the CBA as the invertible mapping

$$\mathrm{MPS}_{j_k}: \qquad \begin{aligned} \mathsf{H}_M^{[r]} &\quad \mapsto \quad \mathsf{H}_k^{[r]} \\ |m_1 \ldots m_r\rangle_M &\quad \mapsto \quad |\Psi_{k,\alpha}^{[r]}\rangle \end{aligned}. \tag{3.48}$$

Since Bethe states are linearly independent thanks to the assumption $u_a \neq u_b$, they span a linear basis of the Hilbert space of the last $k > M$ spins. On the other hand, the last $k-1$ unitaries of ABCs prepare

$$|\Phi_{k,\alpha}^{[r]}\rangle = P_{N-1} \ldots P_{j_k} |m_1 \ldots m_r\rangle_M . \tag{3.49}$$

The set $|\Phi_{k,\alpha}^{[r]}\rangle$ is an orthonormal basis of $\mathsf{H}_k^{[r]}$ because unitary transformations preserve both the orthonormality and the completeness of $|m_1 m_2 \ldots m_r\rangle_M$. Therefore, we can think about the quantum sub-circuit of last $k-1$ unitaries as the unitary mapping

$$\mathrm{ABCs}_{j_k}: \qquad \begin{aligned} \mathsf{H}_M^{[r]} &\quad \mapsto \quad \mathsf{H}_k^{[r]} \\ |m_1 m_2 \ldots m_r\rangle_M &\quad \mapsto \quad |\Phi_{k,\alpha}^{[r]}\rangle \end{aligned}. \tag{3.50}$$

It follows that the matrix $X_j$ performs a change of basis:

$$|\Phi_{k,\alpha}^{[r]}\rangle = \sum_{\beta=1}^{\binom{k}{r}} X_{j_k,\beta\alpha}^{[r]} |\Psi_{k,\beta}^{[r]}\rangle . \tag{3.51}$$

We depict this formula in Figure 16.

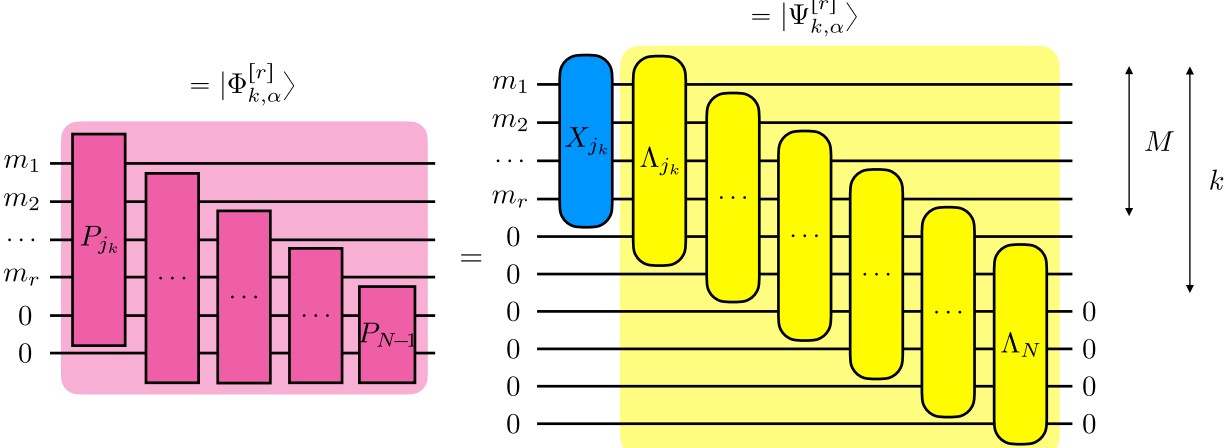

Figure 16: The local gauge transformation $X_{j_k}$ as the change-of-basis matrix that orthonormalizes the set Bethe states with $r$ magnons over $k$ spins.

We choose the change-of-basis matrix in (3.51) to correspond to the standard Gram-Schmidt process, although it is important to notice that $X_j$ is only determined up to unitary rotations. If we use the Gram matrix of Bethe states

$$C_{k,\alpha\beta}^{[r]} = \langle \Psi_{k,\alpha}^{[r]} | \Psi_{k,\beta}^{[r]} \rangle , \tag{3.52}$$

the closed formulae of the matrix elements of $X_j$ are [20]

$$X_{j_k,\alpha\alpha}^{[r]} = \sqrt{\frac{\det_{\alpha-1} C_k^{[r]}}{\det_\alpha C_k^{[r]}}} \;,$$

$$X_{j_k,\alpha\beta}^{[r]} = 0 \;\; \text{if} \;\; \alpha > \beta \;,$$

$$X_{j_k,\alpha\beta}^{[r]} = -\frac{\det_{\beta-1} C_{k,\alpha\to\beta}^{[r]}}{\sqrt{\det_{\beta-1} C_k^{[r]} \det_\beta C_k^{[r]}}} \;\; \text{if} \;\; \alpha < \beta \;,$$

(3.53)

where $\det_\alpha$ denotes the corner principal $\alpha \times \alpha$ minor and $_{\alpha\to\beta}$ denotes the replacement of the $\alpha$-th by the $\beta$-th column. The matrix $X_j$ is upper-triangular. Therefore, $X_j^{-1}$ is upper-triangular as well, and the matrix elements read [20]

$$X_{j_k,\alpha\beta}^{-1[r]} = \frac{\det_\alpha C_{k,\alpha\to\beta}^{[r]}}{\sqrt{\det_{\alpha-1} C_k^{[r]} \det_\alpha C_k^{[r]}}} \;.$$

(3.54)

This matrix provides us with the Cholesky factorization of the Gram matrix (3.52) by construction:

$$C_k^{[r]} = X_{j_k}^{-1[r]\dagger} X_{j_k}^{-1[r]} \;.$$

(3.55)

Table 2 summarizes the features of the matrices in (3.44). The reason for using (3.52) to compute closed fomulae is that we know the entries exactly, thanks to the knowledge of the Bethe states (3.47). However, this knowledge does not mean can evaluate scalar products efficiently in general; rather, the limitation must be taken into account in the numerical computation of the unitaries.

| $1 \leqslant j \leqslant N - M$ | domain | image | # rows $\times$ # columns | formulae |
|---|---|---|---|---|
| $\Lambda_j^{[i,r]}$ | $\mathsf{H}_M^{[r]}$ | $\mathsf{H}_M^{[r-i]}$ | $\binom{M}{r-i} \times \binom{M}{r}$ | (3.34)–(3.36) |
| $X_j^{[r]}$ | $\mathsf{H}_M^{[r]}$ | $\mathsf{H}_M^{[r]}$ | $\binom{M}{r} \times \binom{M}{r}$ | (3.53) |
| $X_{j+1}^{-1[r-i]}$ | $\mathsf{H}_M^{[r-i]}$ | $\mathsf{H}_M^{[r-i]}$ | $\binom{M}{r-i} \times \binom{M}{r}$ | (3.54) |

Table 2: Features the constituents of long unitaries.

It remains to demonstrate unitarity of long unitaries (3.39). In the definition of $\Lambda_j$ in (3.23), we left $\Lambda_j|1\rangle_j$ unspecified because it did not appear in the MPS. This freedom allows us to determine $P_j|1\rangle_j$ at will, which we choose at our best convenience to ensure unitarity. On the other hand, we are left to prove that

$$\langle 0|_j P_{j_k}^\dagger P_{j_k} |0\rangle_j = 1_{2^M} \;,$$

(3.56)

which is equivalent to

$$C_k^{[r]} = \Lambda_{j_k}^{[0,r]\dagger} C_{k-1}^{[r]} \Lambda_{j_k}^{[0,r]} + \Lambda_{j_k}^{[1,r]\dagger} C_{k-1}^{[r-1]} \Lambda_{j_k}^{[1,r]} \;,$$

(3.57)

thanks to the Cholesky factorization (3.55). We offer the proof of (3.57) in Appendix D.

### 3.3.2 Short Unitaries

Our starting point for the construction of short unitaries is the following observation. The matrices $X_{j_k}$ of long unitaries orthonormalize the set of linearly independent Bethe states with $1 \leqslant r \leqslant M$ magnons over the last $M < k \leqslant N$ spins. The construction of $X_{j_k}$ of long gates relies on that the mappings defined by the MPS (3.48) and by the quantum sub-circuit (3.50) are invertible and unitary, respectively. If short unitaries borrowed the mapping defined by the MPS (3.48), it would be non-invertible [7]. Bethe states with $1 \leqslant r \leqslant k$ magnons over the last $1 \leqslant k < M$ spins are not all linearly independent, since they carry every possible subset of $r$ quasi-momenta of $p_{1,j}, \ldots, p_{M,j}$. Furthermore, Bethe states are ill-defined if the number of magnons is greater than the number of spins, that is if $k < r \leqslant M$.

Our strategy to construct the short unitaries consists of two steps. First, we replace the first $M + 1$ tensors $\Lambda_j$ of the MPS (3.25) by the smaller non-unitary tensors

$$\Omega_{j_k} = \Omega_{j_k}(v_{j_k}; u_1, \ldots, u_k) \in \text{End}(\mathsf{H}_{k+1}) . \tag{3.58}$$

The tensor $\Omega_{j_k}$ just acts on $k + 1$ qubits, unlike $\Lambda_{j_k}$. By definition, the MPS of (3.58) must only construct the maximal number of linearly independent Bethe states over $k$ spins out of the computational basis. The number of magnons $r$ of Bethe states is bounded from above by $k$. Next, we define the short unitaries by the local gauge transformation

$$P_{j_k} = X_{j_k+1}^{-1} \Omega_{j_k} X_{j_k} , \quad N - M < j_k \leqslant N - 1 , \tag{3.59}$$

whose matrices follow from the orthonormalization of the set of Bethe states. We depict (3.59) in Figure 17. The new set of matrices are unitary if

$$P_{j_k}^\dagger P_{j_k} = 1_{2^k} . \tag{3.60}$$

We stress the strategy differs from that of [20], which is based on an Ansatz. Our approach is advantageous in that it enables us to demonstrate the construction of short unitaries rigorously.

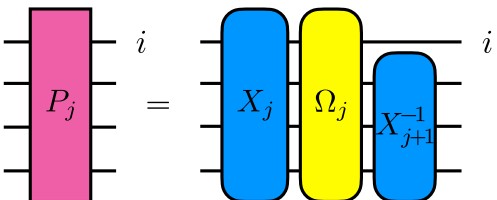

Figure 17: Definition of the last $k + 1$-qubit unitaries out of new non-unitary tensor of the MPS of the CBA by a local gauge transformation. The size of the unitaries is dictated by the number of linearly independent Bethe states over $k$ spins.

The tensor (3.58) breaks into blocks of definite total spin along the $z$-axis by assumption, which in turn split into non-unitary matrices:

$$\Omega_{j_k} = \bigoplus_{r=0}^{k} \Omega_{j_k}^{[r]} , \quad \Omega_{j_k}^{[i,r]} = \langle i|_k \, \Omega_{j_k}^{[r]} , \tag{3.61}$$

---

[7]The MPS of the last $N - M + 1$ tensors $\Lambda_j$ does not suffer from the same problem as it prepares states over $M$ spins. However, we deem $P_{N-M+1}$ to be a short unitary since it acts on $M$ qubits after the elimination of ancillae.

where we recall $\langle i|_k$ belongs to $\mathsf{h}_k$ in (2.1). The number of rows and columns of the non-unitary matrices is

$$\# \text{ rows} \times \# \text{ columns of } \Omega_{j_k}^{[i,r]} = \binom{k-1}{r-i} \times \binom{k}{r} . \tag{3.62}$$

The defining property of the new tensor is that the mapping

$$\text{MPS}_{j_k} : \quad \mathsf{H}_k^{[r]} \quad \mapsto \quad \mathsf{H}_k^{[r]}$$
$$|m_1 \ldots m_r\rangle_k \quad \mapsto \quad |\Psi_{k,\alpha}^{[r]}\rangle = \Omega_N \ldots \Omega_{j_k} |m_1 \ldots m_r\rangle_k \tag{3.63}$$

is invertible. We stress the MPS based on $\Omega_j$ also corresponds to the CBA. Bethe states thus computed carry $r$ quasi-momenta $p_{m_1,\ell} \ldots, p_{m_r,\ell}$, chosen among $p_{1,\ell}, \ldots, p_{k,\ell}$. We depict the equivalence between the MPS of $\Omega_j$ and $\Lambda_j$ in Figure 18.

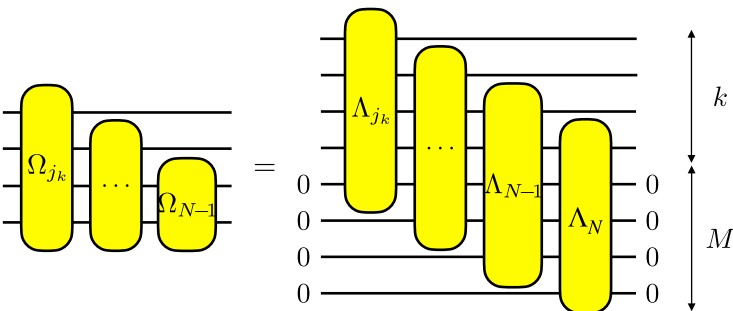

Figure 18: Equivalence between the MPS of the CBA of the $M + 1$-tensor (3.23) and the $k + 1$-tensor (3.58). Despite the difference in the number of input qubits, the equivalence holds due to the linear dependence of Bethe states over $k < M$ spins.

Let us determine the non-unitary tensor. We begin with $j_k = N$, that is $k = 1$. Bethe states on the last spin are just the elements of the computational basis:

$$|\Psi_{1,1}^{[0]}\rangle = |0\rangle , \quad |\Psi_{1,1}^{[1]}\rangle = |1\rangle . \tag{3.64}$$

Therefore,

$$\Omega_N^{[0]}|0\rangle = |0\rangle , \quad \Omega_N^{[1]}|1\rangle = |1\rangle . \tag{3.65}$$

We deduce the last tensor is the identity matrix:

$$\Omega_N = \begin{bmatrix} \Omega_N^{[0,0]} & 0 \\ 0 & \Omega_N^{[1,1]} \end{bmatrix} = \begin{bmatrix} 1 & 0 \\ 0 & 1 \end{bmatrix} . \tag{3.66}$$

Since the last tensor is trivial, the $N$-th unitary is trivial as well, hence its absence in ABCs.

Let $j_k = N - 1$, which implies $k = 2$. According to (3.63), we must have

$$\begin{aligned}
\Omega_{N-1}^{[0]}|00\rangle &= |00\rangle , \\
\Omega_{N-1}^{[1]}|10\rangle &= |10\rangle + x_{1,N-1}|01\rangle , \\
\Omega_{N-1}^{[1]}|01\rangle &= |10\rangle + x_{2,N-1}|01\rangle , \\
\Omega_{N-1}^{[2]}|11\rangle &= (s_{12}x_{2,N-1} - s_{21}x_{1,N-1})|11\rangle .
\end{aligned} \tag{3.67}$$

Therefore,

$$\Omega_{N-1} = \begin{bmatrix} \Omega_{N-1}^{[0,0]} & 0 & 0 & 0 \\ 0 & \Omega_{N-1,1}^{[0,1]} & \Omega_{N-1,2}^{[0,1]} & 0 \\ 0 & \Omega_{N-1,1}^{[1,1]} & \Omega_{N-1,2}^{[1,1]} & 0 \\ 0 & 0 & 0 & \Omega_{N-1}^{[1,1]} \end{bmatrix} = \begin{bmatrix} 1 & 0 & 0 & 0 \\ 0 & x_{2,N-1} & x_{1,N-1} & 0 \\ 0 & 1 & 1 & 0 \\ 0 & 0 & 0 & s_{12}x_{2,N-1} - s_{21}x_{1,N-1} \end{bmatrix}. \quad (3.68)$$

The first non-trivial tensor allows us to compute the remainder.

The computation of the tensor for $N - M < j < N - 1$ is inductive. We detail the derivation in Appendix C. The result is

$$\Omega_{j_k,\alpha\beta}^{[i,r]} = \frac{\det B_{k-1,\alpha\to\beta}^{[r-i]}}{\det B^{[i]}} \ , \quad (3.69)$$

where we used the indexation of Appendix B, and

$$B_{k,\lambda\mu}^{[r]} = {}_k\langle\lambda|\Psi_{k,\mu}^{[r]}\rangle; \quad \left(B_{k,\alpha\to\beta}^{[r-i]}\right)_{\lambda\mu} = \begin{cases} {}_k\langle\lambda|\Psi_{k,\mu}^{[r-i]}\rangle & \text{if} \quad \mu \neq \alpha \\ {}_{k+1}\langle\lambda|\Psi_{k+1,\beta}^{[r]}\rangle & \text{if} \quad i = 0 , \quad \mu = \alpha \\ {}_{k+1}\langle\lambda + \binom{k}{r}|\Psi_{k+1,\beta}^{[r]}\rangle & \text{if} \quad i = 1 , \quad \mu = \alpha \end{cases} \quad (3.70)$$

The matrix $B^{[i]}$ performs change of basis from the computational basis to the set of Bethe states in $\mathsf{H}_{k-1}^{[r-i]}$. The matrix $B_{\beta\to\alpha}^{[i]}$ is the result of the replacement of the $\beta$-th column by the relevant entries of the Bethe state that the MPS (3.63) prepares.

The formula of the non-unitary building blocks of short unitaries is

$$P_j^{[i,r]} = X_{j+1}^{-1[r-i]} \Omega_j^{[i,r]} X_j^{[r]} \ . \quad (3.71)$$

The quantum sub-circuit of the last $k - 1$ short unitaries maps the computational basis to an orthonormal basis of $\mathsf{H}_k^{[r]}$:

$$\text{ABCs}_{j_k} : \qquad \begin{aligned} \mathsf{H}_k^{[r]} & \quad \mapsto \quad & \mathsf{H}_k^{[r]} \\ |m_1 \ldots m_r\rangle_k & \quad \mapsto \quad & |\Phi_{k,\alpha}^{[r]}\rangle = P_{N-1} \ldots P_{j_k}|m_1 \ldots m_r\rangle_k \end{aligned} . \quad (3.72)$$

The matrix $X_{j_k}$ orthonormalizes Bethe states, as we already mentioned. We choose the matrix to be defined by the Gram-Schmidt process again. The entries of the matrix are (3.53) with a suitable adjustment in the ranges of indices. The inverse matrix $X_{j_k}^{-1}$ provides the Cholesky factorization of the Gram matrix (3.52), and the entries of the inverse matrix are (3.54) with the adaptation of the ranges of indices. Table 3 summarizes the features of the matrices in (3.71).

| $N - M < j_k \leqslant N - 1$ | domain | image | # rows $\times$ # columns | formulae |
|---|---|---|---|---|
| $\Omega_{j_k}^{[i,r]}$ | $\mathsf{H}_k^{[r]}$ | $\mathsf{H}_{k-1}^{[r-i]}$ | $\binom{k}{r-i} \times \binom{k}{r}$ | (3.69)–(3.70) |
| $X_{j_k}^{[r]}$ | $\mathsf{H}_k^{[r]}$ | $\mathsf{H}_k^{[r]}$ | $\binom{k}{r} \times \binom{k}{r}$ | (3.53) |
| $X_{j_k+1}^{[r-i]-1}$ | $\mathsf{H}_{k-1}^{[r-i]}$ | $\mathsf{H}_{k-1}^{[r-i]}$ | $\binom{k-1}{r-i} \times \binom{k-1}{r-i}$ | (3.54) |

Table 3: Features of the constituents of short unitaries.

The last step of the construction of short unitaries is the proof of (3.60). According to (3.71) and the Cholesky factorization (3.55), unitarity is equivalent to

$$C_k^{[r]} = \Omega_{j_k}^{[0,r]\dagger} C_{k-1}^{[r]} \Omega_{j_k}^{[0,r]} + \Omega_{j_k}^{[1,r]\dagger} C_{k-1}^{[r-1]} \Omega_{j_k}^{[1,r]} \ . \quad (3.73)$$

We provide the demonstration of (3.73) in Appendix D.

## 4  Conclusions

In this work, we systematized the ABCs of [19, 20], a recent proposal of quantum circuits that prepare arbitrary Bethe states of the standard spin-1/2 XXZ model with periodic boundaries. We demonstrated that the exact unitaries from [20] can alternatively be obtained by performing a basis transformation in the auxiliary space of the ABA. The resulting basis is equivalent to the F-basis, known from the theory of quantum-integrable models.

The key property of the F-basis is that it is symmetric with respect to the exchange of qubits. When applied in the auxiliary space of the ABA, the resulting MPS is invariant with respect to the exchange of the ancillae. A key property of the F-basis is its symmetry with respect to qubit exchange. When applied within the auxiliary space of the ABA, the resulting MPS becomes invariant under the exchange of ancillae. The explicit wave functions of this MPS are scattering plane waves, thus establishing a natural connection to the CBA. As a by-product, the same MPS generates Bethe states with an arbitrary number of magnons simply by changing the initialization in the auxiliary space. While it is relatively straightforward to prove this, we observed that it had not yet been discussed in the F-basis literature. Furthermore, the symmetry of the F-basis clarified how to rigorously eliminate the auxiliary space in the final circuits, so that the ABCs have no ancillae. We showcased our approach with new circuits for preparing the exact Bethe states of the inhomogeneous spin-1/2 XXZ model with periodic boundaries. We believe that the symmetry in qubit exchange in the auxiliary space holds potential for constructing circuits for other quantum-integrable models and could, hopefully, aid in identifying efficient decompositions. Our results open up promising avenues for future research, some of which we briefly discuss below.

First, our ABCs could be applied to models closely related to the inhomogeneous spin-1/2 XXZ model. A straightforward idea is to consider the staggered spin chain [52]. This model can be obtained by choosing alternating inhomogeneities $v_{2j-1} = -iv$ and $v_{2j} = iv$. The Hamiltonian is composed of two- and three-body densities and, much like the homogeneous spin chain, exhibits a rich phase diagram. Moreover, this model holds relevance from another perspective: the alternating spin chain is employed in the "integrable Trotterization" of the XXZ model [53]. This implies that our circuits can construct the exact eigenstates of selected integrable quantum circuits, which are utilized for simulating non-equilibrium dynamics in discrete time. A relatively straightforward generalization of our circuits involves the preparation of Bethe states in the spin-$s$ XXZ models [4, 6], which possess an F-basis in both the quantum and auxiliary spaces [26]. The primary distinction in these models lies in the fact that spins are spin-$s$ qudits, while the ancillae remain qubits, which would affect the elimination of the auxiliary space in the final circuits.

Another spin system that is worth considering is the Richardson-Gaudin (RG) model [3, 54], which realizes doubly degenerate fermions with pair-wise interaction and is quantum integrable [55]. The eigenstates of the RG model are Banerjee-Cooper-Schrieffer states computable by the ABA [56, 57]. The isotropic and anisotropic RG models are related to the "quasi-classical" limits of the transfer matrices of the inhomogeneous XXX [56, 58] and XXZ models [57, 58], respectively; therefore, we expect a similar limit to be applicable to the unitaries of ABCs. The RG model requires to deform the periodic boundary conditions by a diagonal twist, which adds a new layer of complexity to the method without precluding its applicability [59].

A more challenging task is the construction of quantum circuits for other spin chains which can be solved by the Bethe Ansatz. One potential candidate is the solid-on-solid model equivalent to the inhomogeneous spin-1/2 XYZ model [60], which provides the means to construct Bethe

states of this completely anisotropic spin chain [61]. This ice-type model has an F-basis [32], but it is linked to the dynamical YBE [62] rather than the standard YBE (2.9), thus posing a new challenge to ABCs. Even more complex are higher-rank spin chains. While these models do have an F-basis [29–31], it remains uncertain whether it would be beneficial for constructing nested Bethe states. An F-basis in the auxiliary space would be necessary, but achieving such a generalization is not clearly defined. Ultimately, the aim is to construct the nested Bethe states, with the nesting occurring in the auxiliary space. Perhaps the more recent methods discussed in [63, 64] for the ABA in these models could prove advantageous.

## Acknowledgements

R.R. is grateful to Juan Miguel Nieto García for useful discussions. The work of E.L., R.R., G.S., and A.S. has been financially supported by the Spanish Agencia Estatal de Investigación through "Instituto de Física Teórica Centro de Excelencia Severo Ochoa CEX2020-001007-S" and PID2021-127726NB-I00 funded by MCIN/AEI/10.13039/501100011033, by European Regional Development Fund, and the "Centro Superior de Investigaciones Científicas Research Platform on Quantum Technologies PTI-001", by the Ministerio de Economía, Comercio y Empresa through the Estrategia Nacional de Inteligencia Artificial project call "Quantum Spain", and by the European Union through the "Recovery, Transformation and Resilience Plan - NextGenerationEU" within the framework of the "Digital Spain 2025 Agenda". R. R. is supported by the Universidad Complutense de Madrid, Ministerio de Universidades, and the European Union - NextGenerationEU through contract CT18/22. B. P. is supported by the NKFIH excellence grant TKP2021-NKTA-64. The authors are grateful to the organizers of *Exactly Solved Models and Quantum Computing* at Lorenz Center for support and stimulating environment while this work was being completed. R.R. is grateful to the organizers of *Integrable Techniques in Theoretical Physics* at Physikzentrum Bad Honnef for support and stimulating environment and *Recent Advances in Quantum Integrable Systems* at Laboratoire d'Annecy-le-Vieux de Physique Théorique for stimulating environment while this work was being completed.

## A  The Matrix-Product State of the Coordinate Bethe Ansatz

In this appendix, we provide the proofs of claims about the MPS of the CBA we made in the main text. In Subsection A.1, we show the equivalence between the MPS of the CBA of the homogeneous spin chain (3.10) and the MPS of [20]. In Subsection A.2, we demonstrate the equality between both realizations of Bethe states of the inhomogeneous spin chain (3.25): the MPS and the superposition of plane waves. In Subsection A.3, we prove the equivalence between (3.25) and the Bethe wave functions of [49].

### A.1  Equivalent Matrix-Product States

The Bethe state of [20] equals an MPS of the CBA whose tensor appears in (54)–(57) therein. The MPS of [20] follows from (3.10) under the replacement of the scattering amplitudes

$$s_{ab} \mapsto \frac{1}{1 + x_a x_b - 2\Delta x_a} = \frac{\sinh(u_a + \mathrm{i}\gamma)\sinh(u_b + \mathrm{i}\gamma)}{\sinh(\mathrm{i}\gamma)\sinh(u_a - u_b + \mathrm{i}\gamma)} \ , \tag{A.1}$$

and the change of normalization

$$|\Psi_N^{[M]}\rangle \mapsto \left[ \prod_{1 \leqslant b < a \leqslant M} \frac{1}{s_{ba} s_{ab}} \right] |\Psi_N^{[M]}\rangle \,. \tag{A.2}$$

Bethe states in [20] are also explicitly proportional to (3.10). The change of the normalization is

$$|\Psi_N^{[M]}\rangle \mapsto \left[ \prod_{1 \leqslant b < a \leqslant M} \frac{\sinh(\mathrm{i}\gamma) \sinh(u_a - u_b + \mathrm{i}\gamma) \sinh(u_b - u_a + \mathrm{i}\gamma)}{\sinh(u_a + \mathrm{i}\gamma) \sinh(u_b + \mathrm{i}\gamma) \sinh(u_a - u_b)} \right] |\Psi_N^{[M]}\rangle \,. \tag{A.3}$$

## A.2 Inhomogeneous Coordinate Bethe Ansatz

Let us prove the equality between the first and second lines of (3.25). We follow the analogous proof of Appendix B of [20]. We begin with $N = 1$. Formula (3.22) lead us to

$$\sum_{i=0,1} \Lambda_1^i |1\rangle^{\otimes M} |i\rangle = \left[ \prod_{a=1}^{M} x_{a,1} \right] |1\rangle^{\otimes M} |0\rangle + \sum_{a=1}^{M} \left[ \prod_{\substack{b=1 \\ b \neq a}}^{M} s_{ab} x_{b,1} \right] \sigma_a^- |1\rangle^{\otimes M} |1\rangle \,, \tag{A.4}$$

where we borrowed the notation from (3.13) and (3.26). The rightmost state in (A.4) corresponds to the first spin and belongs to the quantum space. The tensor $\Lambda_1$ either preserves the ancillae at $|1\rangle$ and the spin at $|0\rangle$ or flips the state of one single ancilla at a time in exchange for flipping the state of the spin. The fact follows from the commutativity of $\Lambda_1$ with the total spin along the $z$-axis, which encompasses both the $M$ ancillae and the spin.

If $N = 2$, we obtain

$$\sum_{i_j=0,1} \Lambda_2^{i_2} \Lambda_1^{i_1} |1\rangle^{\otimes M} |i_1 i_2\rangle = \left[ \prod_{a=1}^{M} x_{a,1} x_{a,2} \right] |1\rangle^{\otimes M} |00\rangle + \sum_{a=1}^{M} \left[ \prod_{\substack{b=1 \\ b \neq a}}^{M} s_{ab} x_{b,1} x_{b,2} \right] \sigma_a^- |1\rangle^{\otimes M} |10\rangle$$

$$+ \sum_{a=1}^{M} \left[ x_{a,1} \prod_{\substack{b=1 \\ b \neq a}}^{M} s_{ab} x_{b,1} x_{b,2} \right] \sigma_a^- |1\rangle^{\otimes M} |01\rangle + \sum_{a=1}^{M} \sum_{\substack{c=1 \\ c \neq a}}^{M} \left[ \prod_{\substack{b=1 \\ b \neq a}}^{M} s_{ab} x_{b,1} \prod_{\substack{c=1 \\ d \neq a,c}}^{M} s_{cd} x_{d,2} \right] \sigma_c^- \sigma_a^- |1\rangle^{\otimes M} |11\rangle \,. \tag{A.5}$$

The pattern is now clear. If the tensor keeps the $j$-th spin at $|0\rangle$, it yields the product of the quasi-momentum variables $x_{a,j}$ of the ancillae that remain at $|1\rangle$. If the tensor flips the state of the $j$-th spin into $|1\rangle$, the state of one ancilla becomes $|0\rangle$. Let the ancilla be at $a$-th position. The tensor yields the product of scattering amplitudes $s_{ab}$ and quasi-momentum variables $x_{b,j}$ with the ancillae at $|1\rangle$. The eventual projection onto $\langle 0|^{\otimes M}$ in the auxiliary space forces $M$ out $N$ spins to be at $|1\rangle$. The projection and the pattern just explained lead to (3.25).

## A.3 Equivalence with Ovchinnikov's Bethe states

We now demonstrate the equivalence between (3.25) and the Bethe states of [49]. We must perform the non-local gauge transformation of the tensor (3.23):

$$\Lambda_j^i \mapsto W_j^{-1} \Lambda_j^i W_{j-1} \,, \tag{A.6}$$

where

$$W_j = \bigotimes_{a=1}^{M} \left[ \begin{array}{cc} \left[ \prod_{k=1}^{j-1} x_{a,k} \right] \prod_{b=1,b \neq a}^{M} s_{ab} s_{ba} & 0 \\ 0 & \left[ \prod_{k=j+2}^{N} x_{a,k} \right] g_{aj+1} \end{array} \right] \,, \tag{A.7}$$

and we borrowed the notation from (3.13) and (3.26). Non-locality refers to the dependence of the $j$-th matrix $W_j$ on the quasi-momenta $p_{a,j}$ on every position of the spin. (The product by definition equals one if the upper endpoint is smaller than the lower or vice versa.) If we normalize

$$|\Psi_N^{[M]}\rangle \mapsto \left[ \prod_{a=1}^{M} \frac{f_{a1}}{g_{a1}f_{aN}} \prod_{\substack{b=1 \\ b \neq a}}^{M} s_{ab}s_{ba} \right] |\Psi_N^{[M]}\rangle \; , \tag{A.8}$$

to cancel the multiplicative factor induced the non-local gauge transformation and follow the steps of Subsection A.2, we obtain

$$|\Psi_N^{[M]}\rangle = \sum_{1 \leqslant n_1 < ... < n_M \leqslant N} \sum_{\substack{a_1,...,a_M=1 \\ a_p \neq a_q}}^{M} \left[ \prod_{1 \leqslant q < p \leqslant M} \frac{1}{s_{a_p a_q}} \right] \left[ \prod_{p=1}^{M} g_{a_p,n_p} \prod_{j=n_p+1}^{N} x_{a_p,j} \right] |n_1 ... n_M\rangle_N \; . \tag{A.9}$$

Formula (A.9) matches the Bethe states in (13) and (14) of [49].

## B    Indices of Algebraic Bethe Circuits

In this appendix, we define the collective indices for the ordered strings of integers that labels the computational basis of $\mathsf{H}_k^{[r]}$. The labelling of Subsection 3.1 of [20] is the opposite to ours.

Let $1 \leqslant m_1 < ... < m_r \leqslant k$ be the string that labels the state of the computational basis $|i_1 ... i_k\rangle = |m_1 ... m_r\rangle_k$ of $\mathsf{H}_k^{[r]}$. We define the collective index following three steps. First, we rephrase the string in the binary basis as a number in the decimal basis:

$$\chi := \sum_{j=1}^{k} 2^{k-j} i_j = \sum_{j=1}^{k} 2^{k-j} \sum_{p=1}^{r} \delta_{m_p}^{j} \; . \tag{B.1}$$

Note the most significant bit in the string is the first, then the second, and so on. Next, we arrange the numbers in the totally ordered set

$$S = \left( \left\{ \chi = \sum_{j=1}^{k} 2^{k-j} \sum_{p=1}^{r} \delta_{m_p}^{j} : 1 \leqslant m_1 < ... < m_r \leqslant k \right\}, < \right) \; , \tag{B.2}$$

where the order relation $<$ is the standard inequality among integer. Finally, we assign $\alpha$ to the $\alpha$-th element of $S$:

$$|\alpha\rangle_k = |S_\alpha\rangle_k \; , \quad \alpha = 1, ..., \binom{k}{r} \; . \tag{B.3}$$

We illustrate the assignation of collective indices in Table 4.

| $i_1$ | $i_2$ | $i_3$ | $i_4$ | $m_1$ | $m_2$ | $\chi$ | $\alpha$ |
|---|---|---|---|---|---|---|---|
| 0 | 0 | 1 | 1 | 3 | 4 | 3 | 1 |
| 0 | 1 | 0 | 1 | 2 | 4 | 5 | 2 |
| 0 | 1 | 1 | 0 | 2 | 3 | 6 | 3 |
| 1 | 0 | 0 | 1 | 1 | 4 | 9 | 4 |
| 1 | 0 | 1 | 0 | 1 | 3 | 10 | 5 |
| 1 | 1 | 0 | 0 | 1 | 2 | 12 | 6 |

Table 4:   Example of assignation of collective indices for the computational basis.

## C   The Short Tensor

This appendix is devoted to demonstrations around the non-unitary tensor of short unitaries $\Omega_j$ in (3.58). In Subsection C.1, we derive the formula (3.69) for the entries. In Subsection C.2, we prove the equivalence with the Ansatz of [20].

### C.1   Closed Formulae

We derive (3.69) by induction. The base of the induction is (3.68), for which $j = N-1$. Formula (3.69) straightforwardly holds for (3.68). We must prove the inductive step for $N-M < j < N-1$. It follows from (3.63) that

$$
|\Psi_{k,\alpha}^{[r]}\rangle = \Omega_{N-1} \dots \Omega_{j_{k-1}} \Omega_{j_k} |m_1 \dots m_r\rangle_k \tag{C.1}
$$

$$
= \sum_{i=0,1} \sum_{1 \leqslant n_1 < \dots < n_{r-i} \leqslant k-1} |i\rangle \left[ \Omega_{N-1} \dots \Omega_{j_{k-1}} |n_1 \dots n_{r-i}\rangle_{k-1} \right]_{k-1} \langle n_1 \dots n_{r-i} | \Omega_{j_k}^{[i,r]} |m_1 \dots m_r\rangle_k
$$

$$
= |0\rangle \sum_{\beta=1}^{\binom{k-1}{r}} |\Psi_{k-1,\beta}^{[r]}\rangle \Omega_{j_k,\beta\alpha}^{[0,r]} + |1\rangle \sum_{\beta=1}^{\binom{k-1}{r-1}} |\Psi_{k-1,\beta}^{[r-1]}\rangle \Omega_{j_k,\beta\alpha}^{[1,r]} .
$$

Since Bethe states with different number of magnons are linearly independent, two decoupled linear systems for the entries of the tensor (3.58) arise:

$$
\begin{aligned}
\sum_{\beta=1}^{\binom{k-1}{r}} {}_{k-1}\langle \lambda | \Psi_{k-1,\beta}^{[r]}\rangle \Omega_{j_k,\beta\alpha}^{[0,r]} &= {}_k\langle \lambda | \Psi_{k,\alpha}^{[r]}\rangle , \quad \lambda = 1, \dots, \binom{k-1}{r} , \\
\sum_{\beta=1}^{\binom{k-1}{r-1}} {}_{k-1}\langle \lambda | \Psi_{k-1,\beta}^{[r-1]}\rangle \Omega_{j_k,\beta\alpha}^{[1,r]} &= {}_k\langle \lambda + \binom{k-1}{r} | \Psi_{k,\alpha}^{[r]}\rangle , \quad \lambda = 1, \dots, \binom{k-1}{r-1} ,
\end{aligned} \tag{C.2}
$$

where we took into account the indexation of the computational basis of Appendix B. The Cramer's rule provides the solution for both linear systems (C.2) in terms of the change-of-basis matrices between the computational basis and Bethe states (3.70). The result is (3.69).

### C.2   Equality with Other Short Tensors

Reference [20] used another tensor to build short unitaries. Let us demonstrate both tensors are equal. The spin chain of [20] is homogeneous, hence we set $v_j = 0$. According to (54)–(57) and (86) of [20], we must prove

$$
\Omega_{j_k,\alpha\beta}^{[i,r]} = \sum_{\lambda=1}^{\binom{k}{r-i}} L_{j_{k-1},\alpha\lambda}^{[r-i]} \Lambda_{j_k,\lambda\beta}^{[i,r]} , \quad \alpha = 1, \dots, \binom{k-1}{r-i} , \quad \beta = 1, \dots, \binom{k}{r} , \tag{C.3}
$$

where we used the indexation of Appendix B. The entries in left-hand side are (3.69). The right-hand side involves

$$
L_{j_{k-1},\alpha\beta}^{[r-i]} = \frac{\det C_{k-1,\alpha\to\beta}^{[r-i]}}{\det C_{k-1}^{[r-i]}} . \tag{C.4}
$$

The definition of the tensor $\Lambda^i_{j_k}$ is (3.34) and (3.36) under the adaptation of the ranges of the indices. The MPS of $\Lambda^i_{j_k}$ thus defined prepares Bethe states over $k$ spins and $r$ magnons with quasi-momenta in the set $p_1, \ldots, p_k$. Note (C.4) is rectangular:

$$\text{\# rows} \times \text{\# columns of } L^{[r-i]}_{j_{k-1}} = \binom{k-1}{r-i} \times \binom{k}{r-i} . \tag{C.5}$$

The matrix is built from the Gram matrix of Bethe states (3.52). The definition of $C^{[r]}_{k,\alpha\to\beta}$ encompasses scalar of products the maximal set of the linearly independent Bethe states in $\mathsf{H}^{[r]}_k$, with $r$ quasi-momenta in $p_1, \ldots, p_k$, and the linearly dependent Bethe states, with one quasi-momentum $p_{k+1}$ and $r-1$ quasi-momenta in $p_1, \ldots, p_k$.

The first step of the demonstration of (C.3) is

$$C^{[r]}_{k,\alpha\beta} = \langle \Psi^{[r]}_{k,\alpha} | \Psi^{[r]}_{k,\beta} \rangle = \sum_{\lambda=1}^{\binom{k}{r}} \langle \Psi^{[r]}_{k,\alpha} | \lambda \rangle_{kk} \langle \lambda | \Psi^{[r]}_{k,\beta} \rangle . \tag{C.6}$$

The change-of-basis matrix between the computational basis and Bethe states (3.70) Cholesky-factorizes the Gram matrix (3.52):

$$C^{[r]}_k = B^{[r]\dagger}_k B^{[r]}_k . \tag{C.7}$$

Therefore,

$$L^{[r-i]}_{j_{k-1},\alpha\beta} = \frac{\det \widetilde{B}^{[r-i]}_{k-1,\alpha\to\beta}}{\det B^{[r-i]}_{k-1}} , \tag{C.8}$$

where

$$\left( \widetilde{B}^{[r]}_{k,\alpha\to\beta} \right)_{\lambda\mu} = \begin{cases} {}_k\langle \lambda | \Psi^{[r]}_{k,\mu} \rangle = B^{[r]}_{k,\lambda\mu} & \text{if} \quad \mu \neq \alpha \\ {}_k\langle \lambda | \Psi^{[r]}_{k,\beta} \rangle & \text{if} \quad \mu = \alpha \end{cases} . \tag{C.9}$$

The remainder of the proof follows from the recurrence relation among Bethe states.

Let $i = 0$. We use the notation (3.35) in addition to

$$|\lambda\rangle_k = |\ell_1 \ldots \ell_r\rangle_k , \quad j_k \leqslant \ell_1 < \ldots < \ell_r \leqslant N , \tag{C.10}$$

following Appendix B. The labelling also holds with $k$ replaced by $k-1$. We apply (3.34) and (3.69) to (C.3) and obtain

$$\det B^{[r]}_{k-1,\alpha\to\beta} = \det \widetilde{B}^{[r]}_{k-1,\alpha\to\beta} \left[ \prod_{p=1}^r x_{n_p} \right] . \tag{C.11}$$

Formula (3.47) with $v_j = 0$ implies

$${}_k\langle \lambda | \Psi^{[r]}_{k,\beta} \rangle = \sum_{\substack{a_1,\ldots,a_M=1 \\ a_p \neq a_q}}^r \left[ \prod_{1 \leqslant q < p \leqslant M} s_{n_{a_q} n_{a_p}} \right] \left[ \prod_{p=1}^M x^{\ell_p - N + k}_{n_{a_p}} \right] = \left[ \prod_{p=1}^r x_{n_p} \right] {}_{k-1}\langle \lambda | \Psi^{[r]}_{k-1,\beta} \rangle . \tag{C.12}$$

We took into account that

$$N - k + 1 < \ell_1 < \ldots < \ell_r \leqslant N \quad \text{if} \quad \lambda = 1, \ldots, \binom{k-1}{r} . \tag{C.13}$$

Formula (C.11) then follows from then multi-linearity of determinants.

Let $i = 1$. We use the notation (3.37) and

$$|\lambda_1\rangle_{k-1} = |\ell_2 \ldots \ell_r\rangle_{k-1} \ , \quad j_{k-1} \leqslant \ell_2 < \ldots < \ell_r \leqslant N \ . \tag{C.14}$$

We apply (3.34) and (3.69) to (C.3) and obtain

$$\det B^{[r]}_{k-1,\alpha\to\beta} = \sum_{p=1}^{r} \left[ \prod_{\substack{q=1 \\ q\neq p}}^{r} s_{n_p n_q} x_{n_q} \right] \det \widetilde{B}^{[r]}_{k-1,\alpha\to\beta_p} \ . \tag{C.15}$$

We deduce from (3.47) with $v_j = 0$ that

$$_k\langle\lambda|\Psi^{[r]}_{k,\beta}\rangle = \sum_{p=1}^{r} \left[ \prod_{\substack{q=1 \\ q\neq p}}^{r} s_{n_p n_q} x_{n_q} \right] {}_{k-1}\langle\lambda|\Psi^{[r]}_{k-1,\beta_p}\rangle \ , \tag{C.16}$$

where we used

$$N - k + 1 = \ell_1 < \ell_2 < \ldots < \ell_r \leqslant N \quad \text{if} \quad \lambda = \binom{k-1}{r} + 1, \ldots, \binom{k}{r} \ . \tag{C.17}$$

Multi-linearity of determinants implies (C.11). The proof of (C.3) is complete.

# D   Unitarity of Algebraic Bethe Circuits

This appendix is devoted to the demonstration of unitarity of ABCs. In Subsection D.1, we demonstrate (3.57), which implies the unitarity of long unitaries. In Subsection D.2, we demonstrate (3.73), which implies the unitarity of short unitaries.

## D.1   Proof of Unitarity of Long Unitaries

The proof of (3.57) amounts to a direct computation. The entries of (3.57) are

$$C^{[r]}_{k,\alpha\beta} = \left[ \prod_{q=1}^{r} \bar{x}_{m_p,j_k} x_{n_q,j_k} \right] C^{[r]}_{k-1,\alpha\beta} + \sum_{p,q=1}^{r} \left[ \prod_{\substack{x=1 \\ x\neq p}}^{r} \bar{s}_{m_x m_p} \bar{x}_{m_x,j_k} \right] \left[ \prod_{\substack{y=1 \\ y\neq q}}^{r} s_{n_y n_q} x_{n_y,j_k} \right] C^{[r-1]}_{k-1,\alpha_p\beta_q} \ , \tag{D.1}$$

where we labelled Gram matrices by (3.35) and (3.37). Formula (D.1) is the consequence of

$$
\begin{aligned}
C^{[r]}_{k,\alpha\beta} &= \sum_{\substack{j_k\leqslant\ell_1<\ldots<\ell_r\leqslant N}} \sum_{\substack{a_1,\ldots,a_r=1 \\ a_p\neq a_q}}^{r} \sum_{\substack{b_1,\ldots,b_r=1 \\ b_p\neq b_q}}^{r} \left[ \prod_{1\leqslant q<p\leqslant r} \bar{s}_{m_p m_q} s_{n_p n_q} \right] \left[ \prod_{p=1}^{r} \prod_{h=j_k}^{\ell_p-1} \bar{x}_{m_p,h} x_{n_p,h} \right] \\
&= \sum_{\substack{j_{k-1}\leqslant\ell_1<\ldots<\ell_r\leqslant N}} \sum_{\substack{a_1,\ldots,a_r=1 \\ a_p\neq a_q}}^{r} \sum_{\substack{b_1,\ldots,b_r=1 \\ b_p\neq b_q}}^{r} \left[ \prod_{1\leqslant q<p\leqslant r} \bar{s}_{m_p m_q} s_{n_p n_q} \right] \left[ \prod_{p=1}^{r} \prod_{h=j_{k-1}}^{\ell_p-1} \bar{x}_{m_p,h} x_{n_p,h} \right] \\
&\quad \times \left[ \prod_{q=1}^{r} \bar{x}_{m_p,j_k} x_{n_q,j_k} \right] \tag{D.2}
\end{aligned}
$$

$$+ \sum_{\substack{a_1,b_1=1 \\ }}^{r} \sum_{j_{k-1}\leqslant\ell_2<...<\ell_r\leqslant N} \sum_{\substack{a_1,...,a_r=1 \\ a_p\neq a_q \\ a_p\neq a_1}}^{r} \sum_{\substack{b_1,...,b_r=1 \\ b_p\neq b_q \\ b_p\neq b_1}}^{r} \left[\prod_{2\leqslant q<p\leqslant r} \bar{s}_{m_p m_q} s_{n_p n_q}\right] \left[\prod_{p=1}^{r}\prod_{h=j_k}^{\ell_p-1} \bar{x}_{m_p,h} x_{n_p,h}\right]$$

$$\times \left[\prod_{p=2}^{r} \bar{s}_{m_{a_p} m_{a_1}} \bar{x}_{m_{a_p},j_k}\right]\left[\prod_{q=2}^{r} s_{n_{a_q} n_{a_1}} x_{n_{a_q},j_k}\right].$$

## D.2  Proof of Unitarity of Short Unitaries

To proof (3.73), we write the matrix elements

$$C_{k,\alpha\beta}^{[r]} = \sum_{i=0,1} \sum_{\lambda,\mu,\nu=1}^{D_i} \frac{\det B_{k-1,\alpha\to\lambda}^{[r-i]\dagger}}{\det B_{k-1}^{[r-i]\dagger}} \bar{B}_{k-1,\mu\lambda}^{[r-i]} B_{k-1,\mu\nu}^{[r-i]} \frac{\det B_{k-1,\nu\to\beta}^{[r-1]}}{\det B_{k-1}^{[r-i]}} \tag{D.3}$$

where we introduced to alleviate notation

$$D_0 = \binom{k-1}{r}, \quad D_1 = \binom{k-1}{r-1}, \tag{D.4}$$

and we Cholesky-factorized the Gram matrix (3.52) according to (C.7). Since

$$\sum_{\nu=1}^{D_0} B_{k-1,\mu\nu}^{[r]} \frac{\det B_{k-1,\nu\to\beta}^{[r]}}{\det B_{k-1}^{[r]}} = \frac{1}{\det B_{k-1}^{[r]}} \begin{vmatrix} {}_{k-1}\langle 1|\Psi_{k-1,1}^{[r]}\rangle & \cdots & {}_{k-1}\langle 1|\Psi_{k-1,D_0}^{[r]}\rangle & {}_k\langle 1|\Psi_{k,\beta}^{[r]}\rangle \\ \cdots & \cdots & \cdots & \cdots \\ {}_{k-1}\langle D_0|\Psi_{k-1,1}^{[r]}\rangle & \cdots & {}_{k-1}\langle D_0|\Psi_{k-1,D_0}^{[r]}\rangle & {}_k\langle D_0|\Psi_{k,\beta}^{[r]}\rangle \\ {}_{k-1}\langle \mu|\Psi_{k-1,1}^{[r]}\rangle & \cdots & {}_{k-1}\langle \mu|\Psi_{k-1,D_0}^{[r]}\rangle & {}_k\langle \mu|\Psi_{k,\beta}^{[r]}\rangle \end{vmatrix}$$

$$= {}_k\langle \mu|\Psi_{k,\beta}^{[r]}\rangle,$$
$$\tag{D.5}$$

and

$$\sum_{\nu=1}^{D_1} B_{k-1,\mu\nu}^{[r-1]} \frac{\det B_{k-1,\nu\to\beta}^{[r-1]}}{\det B_{k-1}^{[r-1]}} = {}_k\langle \mu + D_0|\Psi_{k,\beta}^{[r]}\rangle, \tag{D.6}$$

thanks to the fact the determinant of a matrix with repeated columns vanishes, we have

$$C_{k,\alpha\beta}^{[r]} = \left[\sum_{\lambda=1}^{D_0} + \sum_{\lambda=D_0+1}^{D_0+D_1}\right] \langle \Psi_{k,\alpha}^{[r]}|\lambda\rangle_{kk}\langle \lambda|\Psi_{k,\beta}^{[r]}\rangle = \langle \Psi_{k,\alpha}^{[r]}|\Psi_{k,\beta}^{[r]}\rangle. \tag{D.7}$$

The proof of (3.73) is thus complete.

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
