# Peer review of "Bethe Ansatz, Quantum Circuits, and the F-basis"

_SciPost Physics_

## Round 1 · Referee Report · Anonymous (Referee 1) · 2025-1-8

Report

Warnings issued while processing user-supplied markup:

  • Inconsistency: plain/Markdown and reStructuredText syntaxes are mixed. Markdown will be used.
    Add "#coerce:reST" or "#coerce:plain" as the first line of your text to force reStructuredText or no markup.
    You may also contact the helpdesk if the formatting is incorrect and you are unable to edit your text.

Ref.: Scipost-202411-00037v1

\vspace{0.1cm}

\textbf{Report} %\newline

\bigskip

In this paper, the authors use the MPS of the CBA to write the unitaries of ABCs for the inhomogeneous XXZ model. They demonstrated that the exact unitaries can alternatively be obtained by performing a basis transformation in the auxiliary space of the ABA.

I believe it deserves to be published in Scipost after the following comments have been appropriately addressed:

\bigskip

  1. \textbf{Reviewer}: Above Eq(3.27), the authors state " (3.25) realizes a Bethe wave functions with M magnons propagating over $N$ spins the inhomogeneous spin chain, we identify it the...". In Section 3.3, they state" Having obtained the MPS of the CBA (3.25), we are in the position to construct ABCs for the inhomogeneous spin chain..." It appears that Eq. (3.25) is crucial. Since the Bethe state should be an eigenstate of the inhomogeneous spin chain, could the authors provide some numerical verification of this equation for small lattice sizes?

%\textbf{Authors}:

\bigskip 2. \textbf{Reviewer}: Above The Table 2, the authors state "However, this knowledge does not mean can evaluate scalar products efficiently in general; rather, the limitation must be taken into account in the numerical computation of the unitaries", Could they elaborate further on what this "limitation" refers to and how it should be addressed? Additionally, the tables should be made clearer. For instance, what do "domain" and "image" mean in Tables 1-3?

%\textbf{Authors}:

\bigskip 3. \textbf{Reviewer}: In Section 3, the number of qubits in which the unitaries act defines two classes: long and short unitaries, and they summarize the properties of long and short unitaries. So, what will happen if the number of spins $N$ becomes infinite? It would be nice to provide some physical interpretation for this.

%\textbf{Authors}:

\bigskip

Please see also the attached pdf file

Attachment

Recommendation

Ask for minor revision

  • validity: -
  • significance: -
  • originality: -
  • clarity: -
  • formatting: -
  • grammar: -

Author:  Roberto Ruiz  on 2025-02-17  [id 5227]

(in reply to Report 1 on 2025-01-08)
Category:
answer to question
correction

Please find attached the response to the referee.

Attachment:

response_scipost_202411_00037v1.pdf

---

## Round 1 · Referee Report · Anonymous (Referee 2) · 2025-5-5

Report

In this paper, the authors demonstrate that the two deterministic formulations of algebraic Bethe circuits proposed in their previous works are related by a change to the F-basis in the auxiliary space. The observation made in this work is interesting and supports its eventual publication in SciPost. However, before I can recommend acceptance, the following points should be addressed:

  1. To my understanding, the Bethe states discussed in this work are the off-shell Bethe states, which are not yet eigenstates of the transfer matrices. There is no discussion of the Bethe ansatz equations, which is another crucial ingredient of the Bethe ansatz. It would therefore be helpful to clarify the distinction between on-shell and off-shell Bethe states and to emphasize that the states considered here are of the latter type.

  2. A point that warrants further explanation is the difference between the two MPS representations of the Bethe states introduced in the authors' previous works. The introduction highlights that one is "numerical" while the other is "analytical." Since the algebraic Bethe ansatz (ABA) is an analytical method for constructing eigenstates (assuming we are dealing with off-shell states), why does the corresponding MPS become "numerical"? Clarifying this distinction would help readers better appreciate the significance of the alternative MPS and the current work.

  3. In ABA, the choice of auxiliary space is arbitrary. For the XXZ spin chain, for instance, the auxiliary space can be in either the fundamental representation or a higher-spin representation. Is a specific choice of auxiliary space required, or is any choice permissible? Addressing this point would be useful, particularly for future applications in constructing spin-s XXZ chains.

  4. Equation (3.2) would benefit from additional explanation—either via an instructive figure or a few more lines of derivation—as it serves as the starting point for the subsequent MPS derivation. The equivalence between (3.1) and (3.2) is not immediately obvious, at least to me (and possibly to other readers as well). Similarly, more clarification on the transition between the second and third equalities in (3.10) would be helpful.

  5. The authors should elaborate on why the key property of the F-basis—namely, its invariance under the exchange of auxiliary states—is useful in transforming the numerical MPS into the analytical MPS. What specific advantage does this property provide?

  6. In the second paragraph of the conclusion, the first two sentences ("A key property of...exchange of the ancilliae") and the following two sentences ("The key property of...exchange of the ancilliae") are nearly identical in meaning. One of these pairs should be removed to avoid redundancy.

Requested changes

See the report

Recommendation

Ask for minor revision

  • validity: top
  • significance: high
  • originality: high
  • clarity: good
  • formatting: good
  • grammar: excellent

Author:  Roberto Ruiz  on 2025-05-17  [id 5490]

(in reply to Report 2 on 2025-05-05)
Category:
answer to question
correction

Please find attached the response to the referee.

Attachment:

response_scipost_202411_00037v1_kxTASX7.pdf

---

## Editorial Decision

resubmitted